# Micrometastasis-derived models enable drug testing for early-stage, high-risk melanoma patients

Kathrin Weidele [iD][1,5], Christian Werno [iD][1,5], Steffi Treitschke [iD][1,5], Catherine Botteron[1],
Martin Hoffmann [iD][1], Sebastian Scheitler[2], Lukas Wöhrl[1], Zbigniew Czyz[2], Giancarlo Feliciello[1],
Florian Weber [iD][3], Adithi Ravikumar Varadarajan[1], Jens Warfsmann [iD][1], Silvia Materna-Reichelt[1],
Marie Katzer [iD][2,4], Laura Schreieder[4], Parvaneh Mohammadi[2], Hedayatollah Hosseini[2],
Kamran Honarnejad[1], Sebastian Haferkamp[4], Melanie Werner-Klein[2,6] & Christoph A Klein [iD][1,2,6 ✉]

## Abstract

**Relapse in melanoma after targeted or immune therapy necessitates the rapid identification of effective alternatives. To address this gap, we investigated whether the timely generation of preclinical models for functional drug testing could reveal additional therapeutic options. Our study focused on: (i) the feasibility of generating in vivo and in vitro models from melanoma lymph node (LN)-derived disseminated cancer cells (DCCs) before relapse, (ii) the implementation of preclinical models to identify therapeutic alternatives, and (iii) the ability to detect patients who could benefit from early functional in vitro drug testing. Successful model generation was significantly associated with DCC quantity, LN origin, and mortality risk. All patient-derived xenograft models were available before patient death and, in 82% of cases, before relapse. Proof-of-concept in vitro drug screening using 315 anticancer drugs identified additional candidates, and coculture of DCCs and LN cells revealed specific T-cell activation and responses to immunotherapy. Our data establish a process for selecting melanoma patients at high risk of progression, enabling the timely generation of patient-derived models to support functionally guided treatment decisions at relapse.**

**Keywords** Disseminated Cancer Cells; Drug Testing; Metastasis; Precision Oncology; Preclinical Models
**Subject Categories** Cancer; Methods & Resources; Skin

## Introduction

During the last five decades, the incidence of melanoma has been steadily rising, leading to 104,960 estimated new cases per year in the United States (Siegel et al, 2025). The vast majority of patients diagnosed with melanoma present with localized disease, which can be treated effectively with surgical resection (Thompson et al, 2009) in most cases. In contrast, distant organ metastasis still represents a major cause of mortality despite the advances in the development of novel therapies (Luke et al, 2017). Hence, almost 8500 melanoma patients are expected to die in the USA from their progressive disease in 2025 (Siegel et al, 2025).

A major clinical breakthrough over the last 15 years has been the validation of effective adjuvant therapies for early-stage melanoma patients after surgical removal of a localized or regional cancer with no evidence of distant metastatic spread (Kobeissi and Tarhini, 2022). Despite widespread adoption of adjuvant therapy, many high-risk patients do not respond to treatment, develop early resistance, or suffer from severe adverse side effects (Gershenwald et al, 2017; Lao et al, 2022). Thus, there remains an urgent need for personalized clinical decision-making and development of individualized treatment strategies for non-metastatic melanoma patients at risk of progression. Currently, this need is primarily addressed by genetic profiling aiming to identify newly acquired or pre-existing, but actionable mutations for targeted therapies. However, across all cancer types, this approach has shown limited success, as therapy resistance often arises from epigenetic and phenotypic alterations which are not readily applied in clinical practice (Garcia-Mayea et al, 2020; Russo et al, 2024).

We have previously shown that quantitative assessment of lymphatic cancer spread by immunocytology enables highly accurate prediction of an unfavorable outcome with higher sensitivity and predictive power than conventional histopathology (Ulmer et al, 2014; Ulmer et al, 2022). Quantification of cancer cells within the LN is easily achieved and can be standardized by applying immunocytology after LN disaggregation. DCCs are counted per million LN cells, which defines the disseminated cancer cell density (DCCD). Strikingly, the logarithm of DCCD is not only linearly correlated with the hazard ratio to die from melanoma, but was also found to be significantly associated with the acquisition of genetic driver alterations (Werner-Klein et al,

[1]Fraunhofer-Institute for Toxicology and Experimental Medicine, Division of Personalized Tumor Therapy, Am Biopark 9, Regensburg 93053, Germany. [2]Experimental Medicine and Therapy Research, University of Regensburg, Franz-Josef-Strauß-Allee 11, Regensburg 93053, Germany. [3]Institute of Pathology, University of Regensburg, Franz-Josef-Strauß-Allee 11, Regensburg 93053, Germany. [4]Department of Dermatology, University Medical Center Regensburg, Franz-Josef-Strauß-Allee 11, Regensburg 93053, Germany. [5]These authors contributed equally: Kathrin Weidele, Christian Werno, Steffi Treitschke. [6]These authors jointly supervised this work: Melanie Werner-Klein, Christoph A Klein. ✉E-mail: christoph.klein@ukr.de

**Table 1. Patient characteristics.**

| Characteristic | Number |
|---|---|
| Patients | 57 |
| Gender | |
| Female | 19 (33.3%) |
| Male | 38 (66.7%) |
| Age (years) | |
| Median | 64 (range 19–79) |
| LN type | |
| SLN | 47 (82.5%) |
| NSLN | 10 (17.5%) |
| Breslow thickness | |
| ≤2 mm | 19 |
| >2–4 mm | 21 |
| >4 mm | 13 |
| n.a. | 4 |
| Ulceration | |
| No | 28 (49.1%) |
| Yes | 24 (42.1%) |
| Not specified | 5 (8.8%) |
| Stage$_{SLN}$[a] | |
| I | 12 (25.5%) |
| II | 12 (25.5%) |
| III | 22 (46.8%) |
| IV | 0 (0%) |
| n.a. | 1 (2.1%) |
| Stage$_{NSLN}$[b] | |
| I | 0 (0%) |
| II | 0 (0%) |
| III | 9 (90%) |
| IV | 1 (10%) |
| Immunocytology[c] | |
| Gp100 negative (DCCD = 0) | 21 (36.8%) |
| Gp100-positive (DCCD > 0) | 36 (63.2%) |
| DCCD | |
| Median | 45 (range 0–900.000) |

*DCCD* disseminated cancer cell density, *LN* lymph node, *n.a.* not available, *SLN* sentinel lymph node, *NSLN* non-sentinel lymph node.

[a]At the time of SLN surgery, including the SLN result.

[b]At the time of NSLN dissection. Of these, five patients had no SLN dissection prior NSLN dissection, five patients underwent SLN dissection before NSLN dissection.

[c]If more than one node per patient was positive, the node with the highest DCCD was taken.

2018), characteristic melanoma phenotypes, and immune response patterns (Guetter et al, 2025). This clearly indicates that DCCs experience selection pressure as they colonize LN and form metastases. In line with this, we had previously noted for individual patients that xenograft formation was closely linked to metastatic colony formation in their LNs (Werner-Klein et al, 2018).

Here, we explored whether xenograft formation and in vitro propagation could be used for the development of functional drug tests that become available before the manifestation of metastasis. We show that xenograft and cell line models of LN-derived DCCs of melanoma patients can be generated not only from late phases of metastatic LN colonization, but also from early phases with low tumor load. We found that functional and genomic features of our DCC-derived models were associated with disease progression. Most importantly for future clinical application, the DCCD and its depending variable, the type of lymphatic origin (i.e., sentinel vs non-sentinel origin; (Ulmer et al, 2018)), were closely linked with model generation and disease progression. Therefore, the DCCD is a quantitative measure for both (i) identifying patients that will likely benefit from functional drug testing and (ii) early selection of LN samples from which patient-derived models for drug testing can be generated with high probability.

## Results

### Generation of melanoma DCC-derived models from LNs

To generate patient-derived melanoma DCC models, we received sentinel lymph nodes (SLNs) and non-sentinel LNs (NSLNs) of a cohort of 57 melanoma patients with varying clinical stages (Table 1). One half of each LN was subjected to routine histopathology, while the other half was disaggregated and analyzed by immunocytology (Ulmer et al, 2014), allowing precise quantification of the tumor load by counting the number of disseminated, gp100-positive melanoma cells per million LN cells (expressed as disseminated cancer cell density, DCCD). To expand melanoma LN-DCCs we employed xenotransplantation followed by in vitro cell line generation (Fig. 1A, approach I) or direct propagation of DCCs in vitro (Fig. 1A, approach II). Patient-derived xenograft (PDX) models were developed either by directly isolating DCCs from LNs based on their expression of the melanoma marker MCSP (Melanoma-Associated Chondroitin Sulfate Proteoglycan) or by pre-enriching LN suspensions under melanosphere culture conditions independent of marker expression (Werner-Klein et al, 2018). The resulting cells and spheres were then transplanted into immunodeficient mice. For cell line (CL) generation, PDX-derived cells (Fig. 1A, approach I) or dissociated LN suspensions or spheres (Fig. 1A, approach II) were cultured under plastic-adherent 2D conditions.

Thirty-six out of 57 patients (63.2%) harbored gp100-positive DCCs in their LNs (Fig. 1A). Of these, 17 (47.2%) successfully formed xenografts when transplanted with varying DCC numbers (Table 2) and 13 out of the 17 PDX models (76.5%) could be expanded after enzymatic dissociation of the PDX tumors and cultured subsequently under 2D plastic-adherent conditions (approach I). Direct in vitro expansion without prior PDX-generation (approach II) was successful for dissociated LN suspensions or their pre-cultured melanospheres in five out of 12 patients (41.6%, Fig. 1A). A detailed overview of the model generation approaches applied to DCC-positive LN samples, including clinical parameters, DCCD, transplanted material, and established models, can be found in Table 2.

Patient-origin and purity of PDX and CL models were confirmed by short tandem repeat analysis (STR) (Fig. EV1A) and flow cytometry for melanoma and stroma-associated markers, respectively (Fig. EV1B).

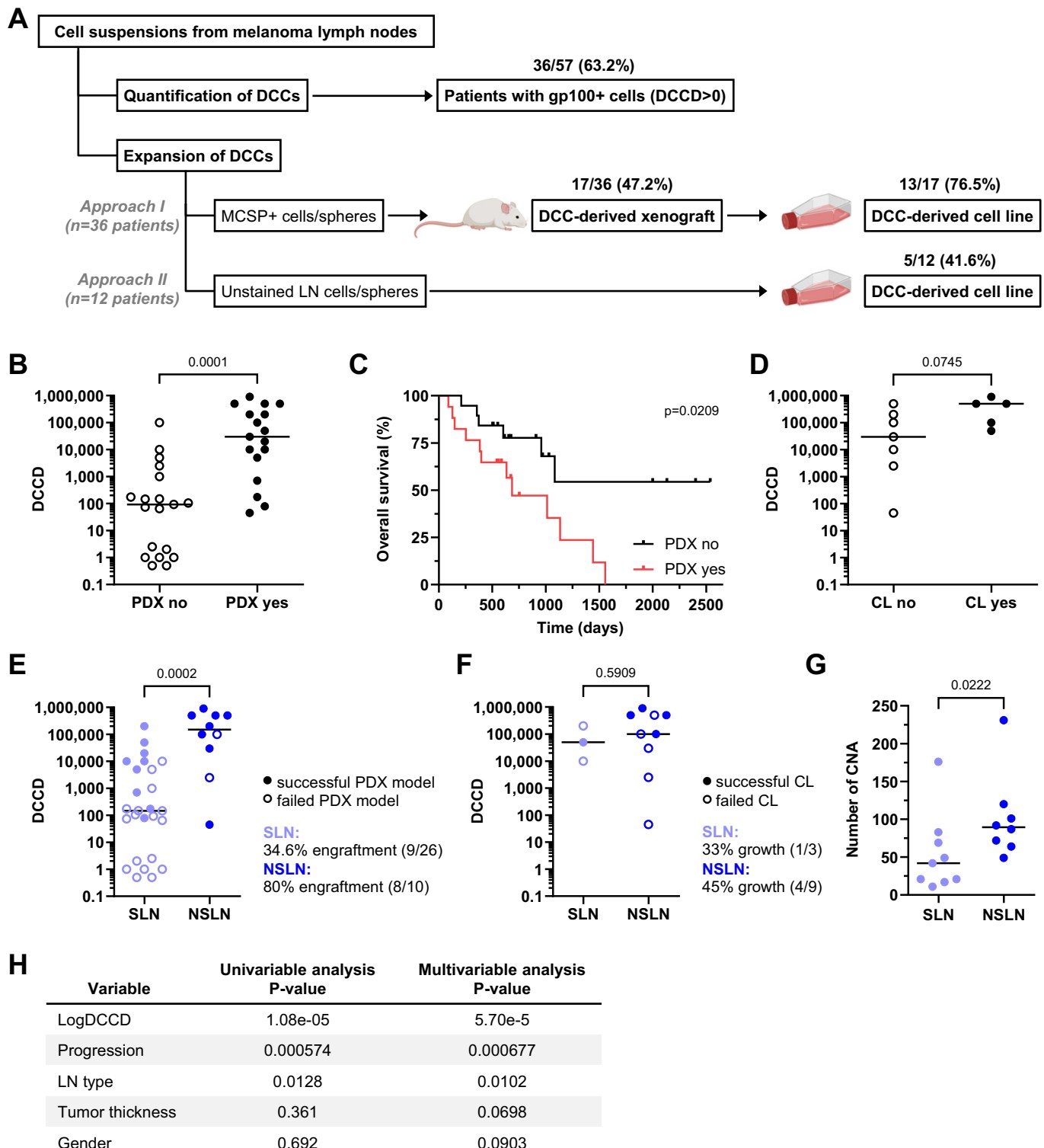

## The DCCD predicts model generation most accurately

To identify factors that influence DCC expansion efficiency, we investigated which sample characteristics were linked with successful model generation. Given the high predictive power of the DCCD for clinical progression and its association with

biological features (Ulmer et al, 2014; Werner-Klein et al, 2018; Guetter et al, 2025), we first investigated whether the DCCD was also associated with successful model generation. Indeed, the DCCD values of successfully generated xenografts were significantly higher than those of samples that failed to engraft ($P < 0.0001$, Mann–Whitney test, Fig. 1B). Moreover, we observed

**Figure 1. DCC model generation is influenced by DCCD and sample origin.**

(A) Schematic overview of the workflow and model creation. LNs from 57 patients with malignant melanoma were disaggregated into cell suspensions. These suspensions were divided into two fractions: one was screened for DCCs using the histogenetic marker gp100, while the other was used for model development. Two approaches were tested: in vivo expansion (by transplanting MCSP+ cells or pre-cultured spheres, approach I) and in vitro expansion (by cultivation of LN cells or disaggregated spheres, approach II). The outcomes and numbers are depicted within the figure. (B) Relationship between DCCD and outcome of in vivo expansion (success in PDX generation) by approach I (PDX yes/no, Mann–Whitney test, two-sided, $n = 36$; ****$P = 0.0001$). Each dot represents an individual patient, the horizontal line indicates the median value for each group. (C) Kaplan–Meier curves of overall survival of patients stratified according to successful PDX generation (PDX yes, $n = 17$ patients) or failure (PDX no, $n = 19$ patients); Log-rank (Mantel–Cox) test, two-sided; $n = 36$ patients; *$P = 0.0209$. (D) Relationship of DCCD and outcome of in vitro expansion and cell line generation by approach II (CL yes/no, Mann–Whitney test, two-sided, $n = 12$ patients; ns (non-significant) $P = 0.0745$). Each dot represents an individual patient, the horizontal line indicates the median value for each group. (E) Relationship of DCCD and outcome of in vivo expansion and PDX generation of SLNs ($n = 26$ patients) vs. NSLNs ($n = 10$ patients) using approach I (unpaired, nonparametric Mann–Whitney test, two-sided; ***$P = 0.0002$). Filled data points indicating patients with successful PDX formation ($n = 17$ patients). Each dot represents an individual patient, the horizontal line indicates the median value for each group. (F) Relationship of DCCD and outcome of in vitro expansion and direct cell line generation of SLNs ($n = 3$ patients) vs. NSLNs ($n = 9$ patients) using approach II (unpaired, nonparametric Mann–Whitney test, two-sided; ns $P = 0.5909$). Filled data points indicate patients with successful CL generation ($n = 5$ patients). Each dot represents an individual patient; the horizontal line indicates the median value for each group. (G) CNA number per DCC model isolated from SLNs vs. NSLNs (unpaired, nonparametric Mann–Whitney test, two-sided, $n = 17$ PDX models; *$P = 0.0222$). Each dot represents an individual patient, the horizontal line indicates the median value for each group. (H) Univariable- and multivariable analysis to determine the association between covariates and PDX establishment. The table represents $P$ values for single variable logistic regression and mean $P$ values across all multivariable logistic regression models for xenograft success that include the indicated variables ($n = 33$ patients). Source data are available online for this figure.

a significantly reduced overall survival (OS) of patients whose samples established PDX compared to patients whose samples failed to engraft ($P = 0.0209$, Log-rank (Mantel–Cox) test, Fig. 1C). For the few CL models established by culturing LN suspensions or spheres in vitro we observed a trend ($P = 0.0745$, Mann–Whitney test, Fig. 1D) that samples with higher DCCD had a higher probability of succeeding. However, the range of the DCCD values in these cases was confined to high DCCD values, which likely blurred the impact of the DCCD.

We had previously shown that the involvement of NSLNs can be predicted with high accuracy based on the DCCD of SLNs (Ulmer et al, 2018), suggesting that NSLN involvement can be considered as a quantitatively extended SLN disease comprising evolutionary progressed cancer cells. Since we had generated models from SLNs at primary treatment and from NSLNs at regional LN adenectomy after local disease recurrence, we compared DCCD values and model generation efficiency of SLN and NSLN samples for in vivo and in vitro expansion. NSLNs showed a significantly higher DCCD than SLNs, both for all samples ($n = 36$, $P = 0.0002$, Mann–Whitney Test, Fig. 1E) and those that successfully engrafted ($n = 17$, $P = 0.0212$, Mann–Whitney Test, Fig. EV2A). Of note, the overall engraftment rate was significantly higher ($P = 0.0248$, Fisher's exact test) in samples derived from NSLNs (80%, 8/10) compared to SLNs (34.6%, 9/26, Fig. 1E). In addition, NSLN samples tended to engraft faster (median: 87.5 days; range: 44–329 days) than SLN samples (median: 248 days; range: 78–338, $P = 0.0274$; Mann–Whitney test, Fig. EV2B). For the 12 CL models successfully generated by culturing LN suspension and melanospheres in vitro, no significant difference was observed between SLN and NSLN samples ($P = 0.59$; Fig. 1F), although more samples are needed to draw a definite conclusion.

We then assessed the molecular evolution of the transplanted cells by analyzing copy number gains and losses (summarized as copy number alterations, CNAs) in the generated xenografts. We found a positive correlation between the DCCD and the number of aberration calls, indicating an increased rate of CNAs with higher DCCD in LNs (Spearman-Rho: 0.5603; $P = 0.0210$; $n = 17$, Fig. EV2C). Consistently, PDXs derived from NSLNs exhibited more CNAs compared to those from SLNs ($P = 0.0222$,

Mann–Whitney test, Fig. 1G). To identify factors affecting the engraftment rate of PDX models, uni- and multivariate analyses were performed (Fig. 1H). We found that the variables logDCCD, progression, and LN type were positively associated with successful PDX generation (univariable analysis), and multivariable analysis confirmed logDCCD as the most significant factor.

## DCC-derived models resemble patient DCCs, regional and distant metastases

We next assessed if the established models retained histomorphological and molecular characteristics, such as CNAs, observed in corresponding patient samples.

All 17 PDX models closely resembled the histomorphologic features of patient-matched LNs (gp100 staining Fig. 2A,C; H&E staining Appendix Fig. S1). As exemplified for patients Mel-27, Mel-39, and Mel-53, the melanoma marker gp100 was consistently expressed in LN samples, corresponding xenografts and in vitro-generated CLs (Fig. 2A). To assess the genomic landscape, we compared the CNA frequency of ex vivo LN-DCCs, PDX models, and CLs. In all tested cases, CNA profiles of matching samples were highly congruent also being confirmed by hierarchical clustering (Figs. 2B,D and EV3A,B; Appendix Fig. S2). For two out of four patients from whom multiple LNs were obtained, we were able to establish models from up to four different LNs per patient (Table 2). In both cases, only minor differences were observed in the CNA patterns across the various sample types (Fig. EV3B).

We next analyzed samples of metastases that were removed up to one year after DCC isolation (48 days for Mel-09, 55 days for Mel-57, and 346 days for Mel-51) and compared them to the matching LN-DCCs, PDX, and CL models (Fig. 2C,D). The DCC-derived models presented similar histological features and gp100 expression as indicated by the H-Score, i.e., the product of staining intensity and percentage of positive cells (Fig. 2C) as well as CNA profiles (Fig. 2D) compared to later arising samples. Remarkably, analysis of distant skin metastases from patient Mel-51 revealed histologic and genomic resemblance with the DCC-derived model generated 346 days prior to the occurrence of distant metastasis.

**Table 2. Patient information of DCCD-positive patients and model generation.**

| Patient ID | Clinical stage[a] | Breslow thickness (mm)[b] | Ulceration | Analyzed LN | DCCD | Expansion approach I | | | | Expansion approach II | | |
|---|---|---|---|---|---|---|---|---|---|---|---|---|
| | | | | | | Transplanted material | PDX | Days to PDX | CL from PDX | Cultured material | CL w/o PDX | CL ID |
| Mel-01 | IIIB | 4.3 | Yes | SLN | 5000 | 10 C | No | | | | | |
| Mel-02 | IIIA | 2.3 | No | SLN | 706 | 7 C | Yes[c] | 248 | Yes | | | Mel-DCC-01 |
| Mel-03 | IIIA | 7 | No | SLN | 174 | 8 C | Yes[c] | 314 | No | | | |
| Mel-04 | IIIA | 2.8 | No | SLN | 105 | 8 C | No | | | | | |
| Mel-05 | IIIC | 5 | Yes | SLN | 5000 | 11 C/5 S | Yes[c] | 338 | No | | | |
| Mel-06 | IIIC | 2.6 | Yes | SLN | 10,000 | 11 C/11 S | Yes[c] | 276 | Yes | | | Mel-DCC-02 |
| Mel-08 | IIA | 3.8 | No | SLN | 1 | 5 S | No | | | | | |
| Mel-09 | IIIB | 3.5 | Yes | SLN | 10,000 | 9 C/10 S | Yes[c] | 238 | Yes | | | Mel-DCC-03 |
| Mel-11 | IIB | 5 | No | SLN | 2 | 6 S | No | | | | | |
| Mel-16 | IIIB | 2.8 | Yes | SLN | 173 | 25 S | No | | | | | |
| Mel-17 | IIIB | 7.5 | Yes | SLN | 1 | 15 S | No | | | | | |
| Mel-23 | IIA | 3.8 | No | SLN | 1 | 10 S | No | | | | | |
| Mel-27 | IIIB | 4.5 | Yes | SLN | 20,000 | 25 S | Yes | 286 | Yes | | | Mel-DCC-04 |
| Mel-28 | IB | 0.9 | No | SLN | 2.5 | 5 S | No | | | | | |
| Mel-33 | IIIA | 1.1 | No | SLN | 64.5 | 5 S | No | | | | | |
| Mel-36 | IIIA | 1.8 | No | SLN | 147.5 | 3 S | No | | | | | |
| Mel-37 | IIIC | 4.5 | Yes | SLN | 200,000 | 200,000 C/40 S | Yes | 88 | No | C | No | |
| Mel-38 | IIID | 6 | Yes | NSLN | 100,000 | 100,000 C/60 S | Yes | 44/119 | Yes | S | Yes | Mel-DCC-05a |
| | | | | NSLN | 100,000 | 200 C/30 S | Yes | 95 | Yes | S | Yes | Mel-DCC-05b |
| | | | | NSLN | 1000 | 10 C/7 S | Yes | 197 | Yes | | | Mel-DCC-05c |
| | | | | NSLN | 100,000 | 45 C/30 S | No | | | | | |
| | | | | NSLN | 1000 | 12 C/30 S | Yes | 207 | No | | | |
| | | | | NSLN | 7 | 20 S | No | | | | | |
| Mel-39 | IIIB | 6 | Yes | SLN | 79 | 55 C | Yes | 247 | Yes | | | Mel-DCC-09 |
| Mel-40 | IIIC | 4 | n.a. | SLN | 50,000 | 200 C/50 S | Yes | 92/118 | Yes | C | Yes | Mel-DCC-06 |
| Mel-41 | IIIA | 2.4 | No | SLN | 93 | 30 S | No | | | | | |
| Mel-42 | IIIB | 2 | No | NSLN | 30,000 | 44 C/100 S | Yes | 329 | No | S | No | |
| Mel-43 | IIIA | 3 | No | SLN | 73.5 | 26 C | No | | | | | |
| Mel-44 | IA | 0.9 | No | SLN | 0.5 | 7 C | No | | | | | |
| Mel-45 | IV | n.a. | n.a. | NSLN | 500,000 | 50 C | Yes | 84 | Yes | C | Yes | Mel-DCC-07 |

**Table 2.** (continued)

| Patient ID | Clinical stage[a] | Breslow thickness (mm)[b] | Ulceration | Analyzed LN | DCCD | Expansion approach I | | | | Expansion approach II | | |
|---|---|---|---|---|---|---|---|---|---|---|---|---|
| | | | | | | Transplanted material | PDX | Days to PDX | CL from PDX | Cultured material | CL w/o PDX | CL ID |
| Mel-46 | II | n.a. | n.a. | NSLN | 2500 | 17 C/ 20 S | No | | | | No | |
| | | | | NSLN | 2500 | 20 S | No | | | | | |
| | | | | NSLN | | 20 S | No | | | | | |
| Mel-47 | IIIC | 3 | No | NSLN | 500,000 | 30 C | No | | | | | |
| | | | | NSLN | 500,000 | 30 C/50 S | Yes | 91 | Yes | C | No | Mel-DCC-08 |
| Mel-48 | IIIB | n.a. | n.a. | NSLN | 45 | 30 C/107 S | Yes | 112/134 | Yes | C | No | Mel-DCC-12 |
| Mel-49 | IIA | 2.5 | No | SLN | 0.5 | 8 S | No | | | | | |
| Mel-51 | IIIC | 3 | No | NSLN | 750,000 | 50 C | Yes | 156 | Yes | | | Mel-DCC-10c |
| | | | | NSLN | 500,000 | 150 C | Yes | 156 | Yes | | | Mel-DCC-10d |
| | | | | NSLN | 750,000 | 100 C/30 S | Yes | 212 | Yes | S | Yes | Mel-DCC-10a |
| | | | | NSLN | 150,000 | 20 S | Yes | 129 | Yes | S | Yes | Mel-DCC-10b |
| Mel-52 | IIIB | 4.4 | Yes | SLN | 10,000 | 40 C | No | | | | No | |
| Mel-53 | IIIC | 4.4 | Yes | NSLN | 900,000 | 300 C/100 S | Yes | 83 | Yes | S | Yes | Mel-DCC-11 |
| | | | | NSLN | 50,000 | 300 C | No | | | | | |
| Mel-54 | IIIC | 9 | Yes | NSLN | 100,000 | 60 C/53 S | No | | | C | No | |
| Mel-55 | IIIA | 3.5 | No | SLN | 146 | 39 C | No | | | | | |
| Mel-56 | II | 2.6 | Yes | SLN | 1000 | 35 C | No | | | | | |
| Mel-57 | IIIC | 2.7 | Yes | NSLN | 200,000 | 300 C/60 S | Yes | 81/90 | Yes | | | Mel-DCC-13 |
| | | | | NSLN | 200,000 | 180 C/60 S | Yes | 98/114 | n.a. | | | |

*Days to PDX* time from receipt of the sample to tumor harvest, *C* cells, *SP* spheres, *CL* cell line, *DCCD* disseminated cancer cell density, *LN* lymph node, *n.a.* not available, *SLN* sentinel lymph node, *NSLN* non-sentinel lymph node, *PDX* patient-derived xenograft, *w/o* without.
[a]At the time of sampling.
[b]At the time of initial diagnosis.
[c]Models already described and published (Werner-Klein et al, 2018).
Underlined values indicate which transplanted material successfully engrafted as PDX.

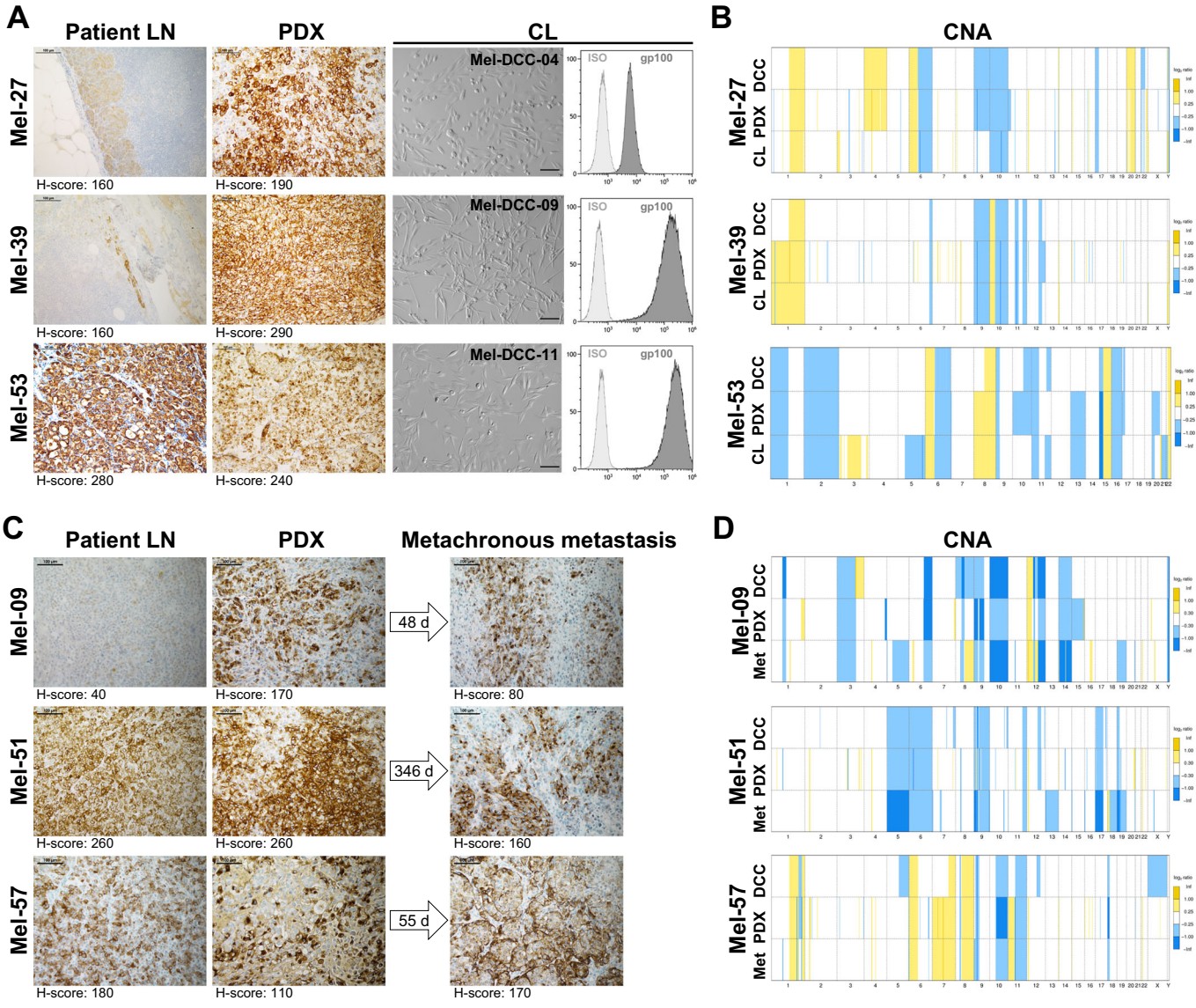

**Figure 2. Analysis and characterization of DCC models and patient samples.**

(A) Histological comparison of patient-derived models (PDXs and CLs) with the corresponding LN specimen of three representative patients. The non-disaggregated half of each patient's LN and corresponding PDX were analyzed by immunohistochemistry (IHC) for gp100 protein expression (brown, scale bar = 100 μm, H-sore is provided under each image). In vitro-generated CLs were imaged by brightfield microscopy (scale bar = 100 μm) and analyzed for gp100 expression by flow cytometry. (B) Comparison of CNA profiles of DCCs, PDXs, and CLs of the three representative patients depicted in (A). Chromosomal gains and losses in DCCs, DCC-derived PDXs, and in vitro-generated CLs are depicted in yellow and in blue, respectively. For Mel-53, chromosomes X and Y were omitted because these were unavailable for the CL analyzed by low-pass sequencing. (C) IHC of patient-derived LN samples for gp100, with the corresponding PDX tumors and metastases that developed after the indicated time period following LN removal. (scale bar = 100 μm, H-score is provided under each image). (D) Comparison of CNA profiles of LN-DCCs, PDXs, and metastases of the three patients depicted in (C). Color-coding as in (B). Source data are available online for this figure.

In vitro selection is known to affect reproducibility of cancer cell line experiments (Quevedo et al, 2020; Liu et al, 2019). Therefore, we assessed the genomic stability of two PDX and two CL models over four in vivo or 20 in vitro passages. We observed that PDX models remained genomically stable with only minor changes (Fig. EV3C, upper panel). Likewise, in vitro expanded CLs displayed comparable genomic profiles after transfer from PDX to plastic-adherent conditions, and the profiles remained stable during cultivation for at least 20 passages (Fig. EV3C, lower panel).

Cell lines generated in parallel by approaches I and II displayed similar key features, including melanoma surface-marker expression and CNA profiles (Appendix Fig. S3).

In summary, we successfully established 17 PDX and 13 CL models from 17 out of 36 patients. These models exhibited consistent histomorphological characteristics and maintained genomic stability, accurately reflecting the genomic profile and CNA pattern of LN metastasis. This stability was preserved across multiple in vitro and in vivo passages.

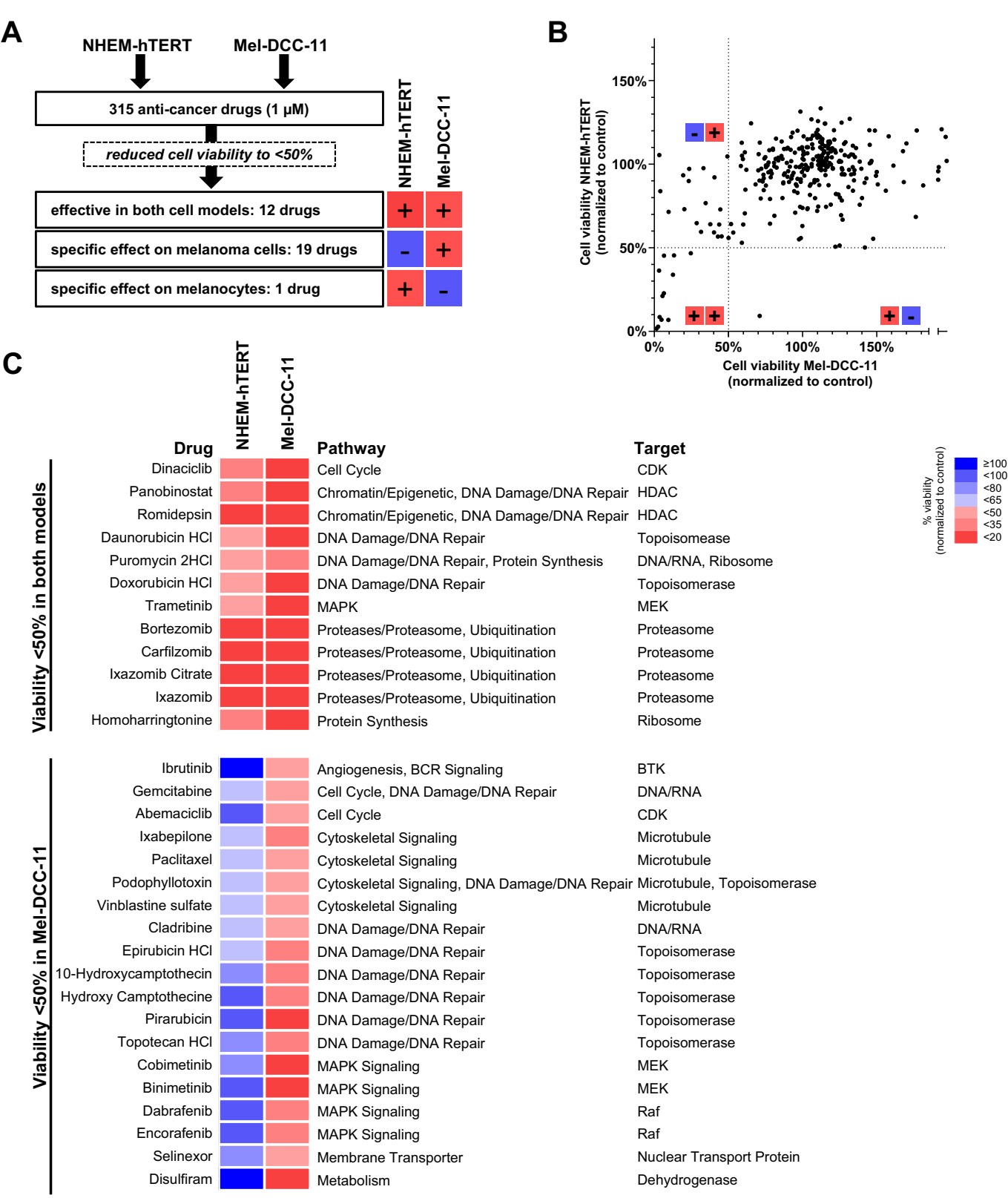

**Figure 3.  Drug screening of 315 anti-cancer drugs reveals melanoma-specific drug responses.**

(A) Schematic overview of drug screening comparing non-malignant melanocytes (NHEM-hTERT) and the *BRAF* V600E-mutated Mel-DCC-11 cell line. The number of drugs that reduced cell viability by more than 50% is indicated, with comparisons selected based on viability values shown in (B). (B) Viability values of NHEM-hTERT plotted against the viability values of Mel-DCC-11 for all tested 315 compounds at 1 µM. Each dot represents the mean value of two biological replicates. The dotted lines separate the toxic (+/+), Mel-DCC-11-specific (−/+), and Mel-DCC-unspecific (+/−) drugs, which reduce the viability to less than 50% in each group. (C) Heatmap of viability values for top-scoring drugs affecting NHEM-hTERT and Mel-DTC-11 (upper panel, $n = 12$ drugs) or selectively inhibiting Mel-DCC-11 viability by more than 50% (lower panel, $n = 19$ drugs). The mean viability values of two biological replicates are shown. Source data are available online for this figure.

**Table 3.  DCC-derived cell lines.**

| CL ID | Origin | Doubling time (h) | Mutational status (VAF) | | Drug response (IC$_{50}$) | |
| --- | --- | --- | --- | --- | --- | --- |
| | | | **BRAF** | **NRAS** | **Vemurafenib** | **Binimetinib** |
| Mel-DCC-01 | PDX | 57.5 | wt | Q61K (50%) | No impact | 15 nM |
| Mel-DCC-02 | PDX | 118 | V600E (65%) | wt | 0.88 µM | 26.5 nM |
| Mel-DCC-03 | PDX | 55.4 | V600E (50%) | wt | 1.66 µM | n.a. |
| Mel-DCC-04 | PDX | 41.4 | wt | Q61R (100%) | No impact | No impact |
| Mel-DCC-05a | LN-DCC | 48.3 | wt | Q61K (100%) | No impact | 75 nM |
| Mel-DCC-06 | LN-DCC | 55.9 | V600K (100%) | wt | 0.73 µM | n.a. |
| Mel-DCC-07 | LN-DCC | 59.8 | wt | T58I (50%) | No impact | No impact |
| Mel-DCC-08 | PDX | 72.3 | wt | G12C (60%) | n.a. | 110 nM |
| Mel-DCC-09 | PDX | 117.9 | wt | Q61R (50%) | No impact | 51 nM |
| Mel-DCC-10a | LN-DCC | 55.2 | wt | Q61K (100%) | n.a. | 28.4 nM |
| Mel-DCC-11 | LN-DCC | 64.9 | V600E (100%) | wt | 0.75 µM | 43 nM |
| Mel-DCC-12 | PDX | 49.2 | wt | wt | No impact | 46 nM |
| Mel-DCC-13 | PDX | 66.6 | V600K (100%) | wt | 1.05 µM | 56 nM |

*CL* cell line, *DCC* disseminated cancer cell, *LN* lymph node, *n.a.* not analyzed, *PDX* patient-derived xenograft, *VAF* variant allele frequency, *wt* wild type.

## Compound screening reveals drug vulnerabilities specific to melanoma

To demonstrate proof of concept, we screened a library of 315 anti-cancer drugs at a single concentration of 1 µM using a representative CL model (Mel-DCC-11, derived from patient Mel-53). As control, we included a cell line model generated from non-malignant immortalized melanocytes (normal human epidermal melanocytes; NHEM-hTERT) to identify screening hits that were non-specific to malignant cells (Fig. 3A). Initially, 12 out of 315 drugs were flagged as non-specific as they reduced the viability of both Mel-DCC-11 and NHEM-hTERT to less than 50%; these included proteasome inhibitors, HDAC inhibitors or drugs involved in DNA/RNA synthesis (Fig. 3B,C, upper panel). One drug selectively reduced the viability of NHEM-hTERT. Ultimately, we found 19 DCC-specific drugs that selectively reduced the viability of Mel-DCC-11 cells to less than 50% (Fig. 3B,C, lower panel). These drugs targeted several key mechanisms, including cell cyle, DNA damage/DNA repair, cytoskeletal signaling (e.g., CDK inhibitors, microtubule disruptors, and topoisomerase inhibitors like Abemaciclib, Ixabepilone, Pirarubicin, Camptothecine, or Topotecan) as well as MAPK signaling (Fig. 3C, lower panel). In particular, the MEK/RAF inhibitors Cobimetinib, Binimetinib, Dabrafenib and Encorafenib exhibited high efficacy in Mel-DCC-11, reducing cell viability to less than 35%. Remarkably, the specific compounds comprised not only well-established melanoma drugs but also drugs with no prior reported relevance to melanoma

treatment. Disulfiram, a drug currently used to treat chronic alcoholism but known to exhibit anti-cancer activity (Iljin et al, 2009; Chen et al, 2006), was highly effective in inducing tumor cell death, reducing cell viability to less than 1%.

## DCC models enable personalized testing of response to targeted treatments

The response of Mel-DCC-11 to MEK inhibitors was expected, as abnormal activation of the RAS/RAF/MEK/ERK pathway occurs in more than 80% of melanomas. MAPK pathway inhibition has been shown to be highly effective in treating metastatic melanoma. However, most metastatic patients eventually develop resistance mechanisms, leading to disease progression within six to eight months of treatment, while some patients exhibit intrinsic resistance, preventing initial treatment success (Angelis and Kanavos, 2016).

Five of our 13 (38.5%) in vitro CL models displayed mutations in *BRAF*, including V600E and V600K, while seven models carried *NRAS* mutations (53.8%), including Q61K, Q61R, T58I and G12C. One cell line showed neither *BRAF* nor *NRAS* hotspot mutations (Table 3). Based on these mutations, we determined IC$_{50}$ values for Vemurafenib (PLX4032), a BRAF kinase inhibitor, and Binimetinib (MEK162, ARRY-162), a MEK kinase inhibitor (data summarized in Table 3, dose–response data for all tested CL models are shown in Fig. EV4). As expected, only BRAF V600-mutated CL models responded to Vemurafenib (Fig. 4A, upper graph), while inhibition

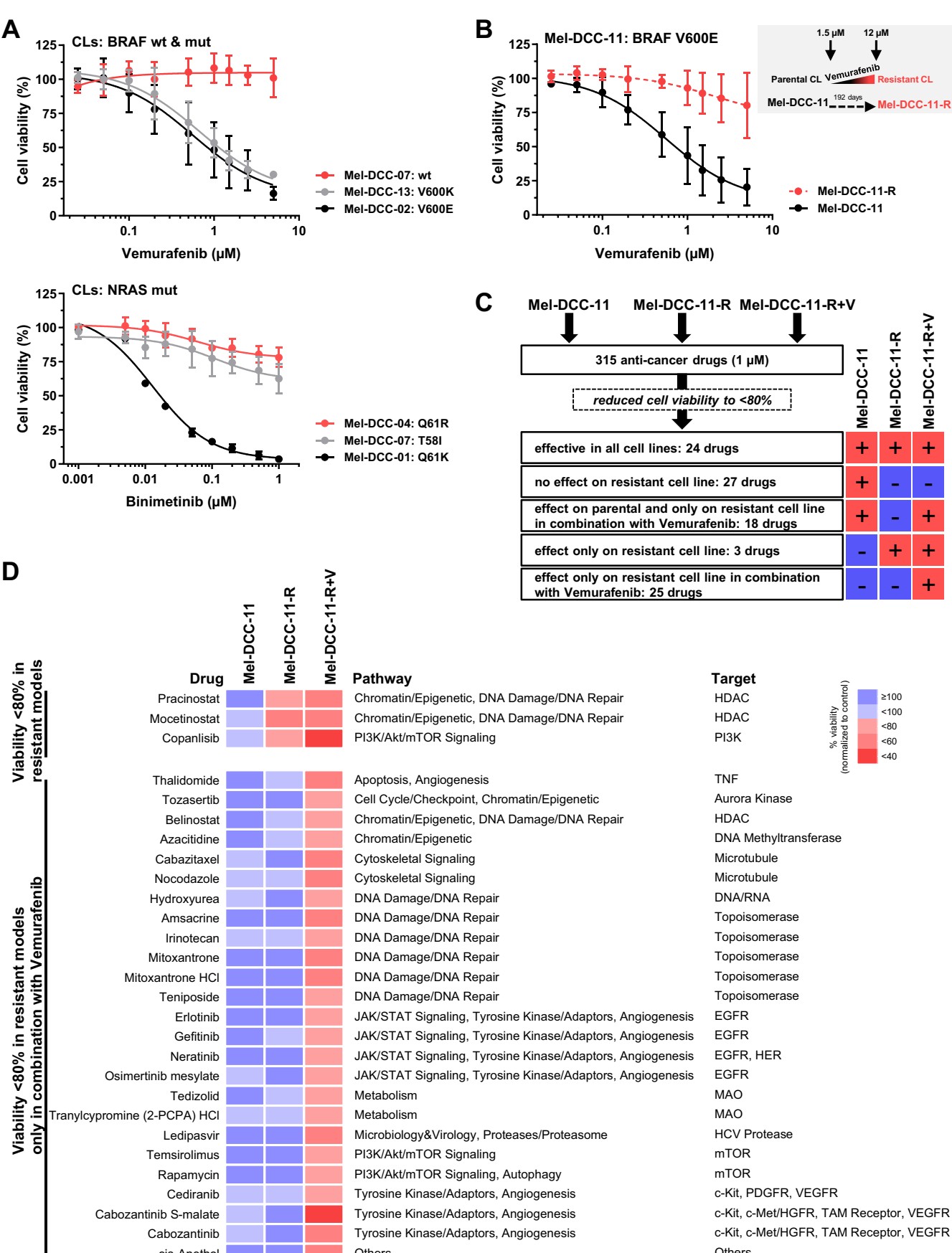

**Figure 4. Response to targeted treatments and identification of pathways mediating resistance.**

(A) Upper graph: Vemurafinib treatment of Mel-DCC CLs. Cells were incubated with doses ranging from 0.025 µM to 5 µM Vemurafenib for 5 days. Cell viability is shown for BRAF wt Mel-DCC-07 (red, $n = 4$), BRAF V600K-mutated Mel-DCC-13 (gray, $n = 4$), and BRAF V600E-mutated Mel-DCC-02 (black, $n = 5$). Lower graph: Binimetinib treatment of Mel-DCC CLs. Cells were incubated with Binimetinib at doses ranging from 0.001 µM to 1 µM for 5 days. Cell viability is shown for NRAS Q61R-mutated Mel-DCC-04 (red, $n = 4$), NRAS T58I-mutated Mel-DCC-07 (gray, $n = 6$), and NRAS Q61K-mutated Mel-DCC-01 (black, $n = 4$). Each dot represents the mean value ± SD of biological replicates. (B) Generation of a Vemurafenib-resistant BRAF-mutated melanoma cell line (Mel-DCC-11-R). Resistance was generated through stepwise exposure to increasing concentrations of Vemurafenib over the indicated timeframe. Sensitivity of Mel-DCC-11 (black, $n = 3$) vs. Mel-DCC-11-R (red, $n = 5$) to Vemurafenib is shown. Each dot represents the mean value ± SD of biological replicates. (C) Outcome of experimental drug testing with 315 anti-cancer drugs on BRAF V600E-mutated Mel-DCC-11 and Vemurafenib-resistant Mel-DCC-11-R, alone or in combination with 8 µM Vemurafenib (Mel-DCC-11-R + V). The number of drugs that reduce cell viability to less than 80% is indicated. (D) Heatmap showing the drug-induced reduction of the viability in the Vemurafenib-restistant CL, screened in the presence (Mel-DCC-11-R + V) or absence (Mel-DCC-11-R) of Vemurafenib, alongside the parental Vemurafenib-sensitive Mel-DCC-11, screened without Vemurafenib. The mean viability of two biological replicates is shown. Source data are available online for this figure.

of MEK signaling was effective in nearly all CL models except for two, Mel-DCC-04 (NRAS Q61R mutation) and Mel-DCC-07 (NRAS T58I mutation; Fig. 4A, lower graph). Both responded only weakly to Binimetinib, suggesting the presence of a pre-existing, intrinsic drug resistance.

To model acquired resistance, we exposed two *BRAF*- and two *NRAS*-mutated CL models to increasing concentrations of Vemurafenib or Binimetinib, successfully generating resistant CL models for *BRAF*-mutant Mel-DCC-11 (Fig. 4B) and Mel-DCC-03 (Fig. EV5) and for the *NRAS*-mutant Mel-DCC-05 and Mel-DCC-10 (Fig. EV5).

In addition, we performed targeted-panel sequencing covering the coding sequences of 410 cancer-associated target genes exemplarily on Mel-DCC-11 (parental, sensitive to BRAF inhibitor) and its Vemurafenib-resistant derivative Mel-DCC-11-R) to identify resistance-causing mutations. Although mutations of unclear significance were detected in both models, only the BRAF V600E mutation was recognized as known oncogenic (Table EV1), indicating resistance may arise from either non-genetic factors or other, yet unknown, genetic causes. This prompted us to conduct a functional investigation into resistance. To achieve this, we performed a drug screen using the library of 315 anti-cancer drugs side-by-side on both the parental CL (Mel-DCC-11) and the Vemurafenib-resistant CL (Mel-DCC-11-R), with the latter also in combination with Vemurafenib (Mel-DCC-11-R + V). Our primary goal was to identify compounds universally active across the tested models and to identify drugs that are effective alone or that selectively restore sensitivity to Vemurafenib when used in combination (Fig. 4C). We identified 24 drugs that reduced viability by at least 20% across all three tested models, including 12 drugs that were also effective in NHEM-hTERT cells. In addition, 27 drugs exhibited no efficacy in the resistant models compared to Mel-DCC-11, questioning their suitability as second-line treatment following the emergence of BRAF inhibitor resistance. Interestingly, three compounds, a PI3K and two HDAC inhibitors, specifically reduced the viability of resistant cells, but had no effect on the parental Mel-DCC-11 cells (Fig. 4D, upper panel). This effect was even more pronounced when these inhibitors were combined with Vemurafenib (Mel-DCC-11-R + V). Furthermore, we identified several drugs and targets that restored sensitivity to Vemurafenib. Among the top 25 compounds that reduced the viability of resistant Mel-DCC-11 cells in the presence of Vemurafenib, we observed a preponderance of topoisomerase inhibitors as well as tyrosine kinase inhibitors (targeting EGFR, VEGFR, c-kit or c-Met; Fig. 4D, lower panel).

In summary, our workflow enables the identification of patient-specific vulnerabilities to targeted therapies, as well as cases of de novo resistance, offering a valuable opportunity to discover resistance-breaking drug combinations for potential second-line treatment.

## DCC models enable analysis of T-cell activation and immune checkpoint inhibition

Immune checkpoint inhibitors (ICIs) have transformed melanoma therapy. To test whether T-cell activation and immune checkpoint inhibition can be assessed in vitro using our DCC models, we established cocultures of Mel-DCC CLs with LN cell suspensions. Due to the unavailability of autologous biobanked LN samples, LN suspensions from allogeneic DCC-negative LNs were used (LN-01, -02, and -03). The LN suspensions were predominantly composed of CD4$^+$ T cells (49–72%), B cells (14–43%), and CD8$^+$ T cells (7–13%), with NK cells below the detection threshold (<2%) (Fig. 5A). CD8$^+$ T cells, the primary mediators of antigen-specific tumor cell killing and the core target of many effective immunotherapies, were mainly of naive (31–59%) or central memory (CM) phenotype (38–61%) (Fig. 5B). They did not show signs of exhaustion as indicated by the low percentage of PD-1$^+$Tim-3$^+$ T cells (Fig. 5C), very similar to levels observed in LN of non-tumor patients (Guetter et al, 2025).

To analyze whether LN-derived CD8$^+$ T cells can be functionally activated, we first stimulated LN suspensions with anti-CD3/CD28, both with and without exogenous IL-2. T-cell activation and proliferation were indicated by increased frequencies of CD25$^+$, PD-1$^+$, and Tim-3$^+$ T cells, along with a higher frequency of Ki-67$^+$ T cells at day 7 (Fig. 5D). We then tested if CD8$^+$ T cells can also be activated by coculture with PD-L1–negative Mel-DCC-01 cells (Fig. EV1B). Indeed, coculture with Mel-DCC-01 in the presence of IL-2 provoked a significant increase in CD25$^+$, Ki-67$^+$, and a non-significant trend towards elevated Tim-3$^+$, and PD-1$^+$ frequency of CD8$^+$ T cells compared to IL-2 treatment alone (Fig. 5D).

Next, we cocultured PD-L1-expressing Mel-DCC-11 cells (Fig. EV1B) with LN cell suspensions to test if the presence of PD-L1 and/or the addition of ICIs will influence the alloreactive T-cell activation. Despite a high LN donor variation in the IL-2 control, a clear trend was visible that coculture with Mel-DCC-11 in the presence of the isotype control failed to induce CD8$^+$ T-cell activation and proliferation, as indicated by the unchanged frequency of CD25$^+$ and PD-1$^+$ CD8 T cells, along with a reduced frequency of Ki-67$^+$ and Tim3$^+$ CD8 T cells (Fig. 5E). To evaluate

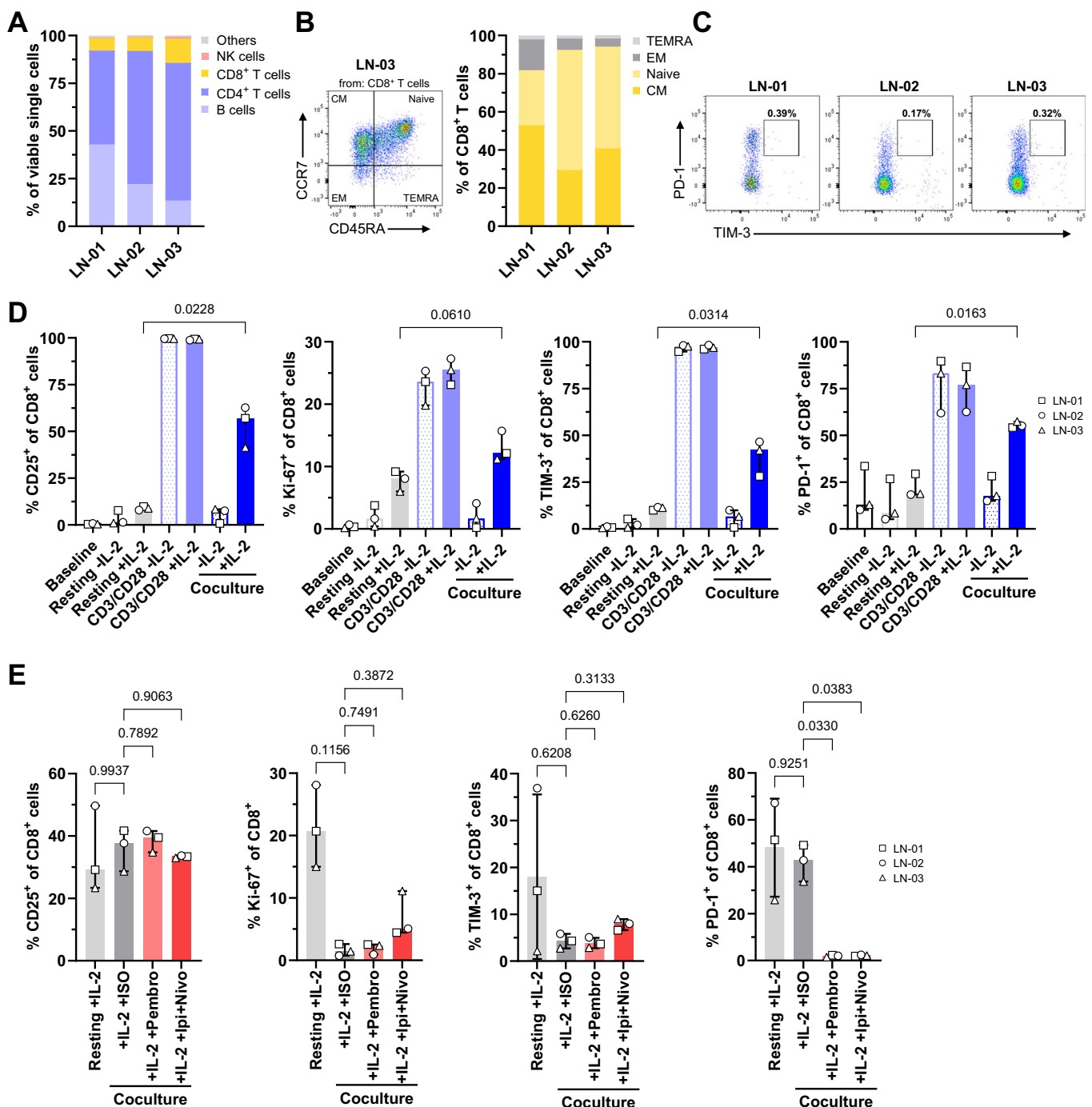

**Figure 5. Immune modulation in cocultures of Mel-DCC and LN cells.**

(A) Flow cytometric analysis of immune cells in DCC-negative LNs of three melanoma patients (LN-01, -02, -03). Relative frequencies of CD4+ and CD8+ T cells, B cells, and NK cells are shown. (B) Flow cytometric assessment of CD8+ T-cell subtypes in LN-01, -02, and -03. Left graph: Representative pseudocolor plot showing the fraction of naive, central memory (CM), effector memory (EM), and terminal effector memory (TEMRA) CD8+ T cells in LN-03. Right graph: Bar graph summarizing relative T-cell subset frequencies across donors ($n = 3$). (C) Relative frequency of PD-1+TIM-3+ T cells in LN 01-03 analyzed by flow cytometry. (D) Flow cytometric analysis of the relative frequencies of CD25+, Ki-67+, TIM-3+, and PD-1+ CD8+ T cells in LN-01, -02, and -03 at baseline (immediately after thawing) and after 7 days of culture under different conditions. LN cells were cultured in medium alone (resting -IL-2), with IL-2 (resting +IL-2), with anti-CD3/CD28 beads (±IL-2) or together with Mel-DCC-01 (coculture ± IL-2). Each dot represents one donor, the median with range (min to max) is shown. $P$ values were determined using a one-way ANOVA followed by Šídák's multiple comparisons test (*$P = 0.0136$–$0.0314$, ns (non-significant) $P = 0.0610$). (E) Flow cytometric analysis of LN-01, -02, -03 cultured with IL-2 alone (resting +IL-2) or together with Mel-DCC-11 (coculture) in the presence of IL-2 (+IL2 + ISO) or checkpoint inhibitors Pembrolizumab (+IL2 +Pembro, 10 μg/ml), Ipilimumab + Nivolumab (+IL2 +Ipi +Nivo, each 10 μg/ml). Each dot represents one donor, the median with range (min to max) is shown. $P$ values were determined using a one-way ANOVA followed by Šídák's multiple comparisons test (*$P = 0.0330$–$0.0383$; ns $P = 0.1156$–$0.9937$). Source data are available online for this figure.

whether T-cell activation can be restored by ICIs, we additionally treated Mel-DCC-11/LN cocultures with pembrolizumab or ipilimumab plus nivolumab (Fig. 5E). Notably, only the combination therapy increased slightly the relative frequency of Ki-67$^+$ and Tim-3$^+$ CD8$^+$ T cells, but not CD25$^+$ CD8$^+$ T cells, indicating only partial restoration of T-cell activation and potential involvement of additional checkpoint mechanisms (Guetter et al, 2025). However, due to variation in the effect between individual donors, the changes did not reach statistical significance. As expected, PD-1$^+$ CD8$^+$ T cells were barely detectable after treatment, consistent with full receptor occupancy by pembrolizumab and nivolumab. In summary, these findings demonstrate that allogeneic Mel-DCC/LN cocultures provide a feasible and informative platform to assess the potential of patient-derived CLs to suppress T-cell activation and to measure responsiveness to immune checkpoint inhibition in vitro.

### Timely model generation for clinical application in high-risk melanoma patients

For patient-derived models to be clinically useful for personalized drug testing, they must enable identification of effective drugs within the patient's lifetime, ideally before metastasis becomes clinically apparent. To evaluate the feasibility of this approach, we assessed the time needed for model generation in relation to disease progression. We observed a correlation between the time to PDX model establishment and the time to disease progression in patients, with faster progressing melanomas resulting in earlier model generation ($r = 0.24$; Fig. 6A).

Notably, all models were generated before the respective patient's death (Fig. 6B). The median time between PDX generation and patients' death or last follow-up was 607 days for SLNs (range 14–1165 days) and 426 days for NSLNs (range 9–589 days). In comparison, the median time required from LN excision to the availability of the first PDX models was 129 days (range: 44–338 days). In addition, in more than 82% of cases (14/17) a xenograft could be established before disease progression was diagnosed. Strikingly, 6/9 models derived from SLN DCCs and all eight models derived from NSLN DCCs could be generated even prior to diagnosis of distant metastatic progression (Fig. 6C). Given that in vitro expansion and high-throughput drug testing may take up to 100 days, we could have identified personalized therapies for 59% of patients (10/17) before the onset of distant metastases (Fig. 6C, dotted line). To identify patients with a high likelihood of successful preclinical model generation, stratification based on their LN DCCD can be applied. In a previous study, we identified a DCCD of ≥100 as a threshold for the onset of metastatic colonization and xenograft formation (Werner-Klein et al, 2018). Consistently, our current analysis shows that samples from patients with a DCCD ≥ 100 had a significantly higher overall engraftment rate (15/24; 62.5%) compared to those with lower DCCD values (2/12; 16.7%, $P = 0.0140$, Fisher's exact test; Fig. 6D). This finding supports the use of the DCCD ≥ 100 threshold as a criterion for selecting patients for model generation at the time of primary treatment. We therefore examined the prevalence and clinical course of patients with a DCCD ≥ 100 in a recently reported larger cohort of 380 patients (Guetter et al, 2025). As expected, these patients experienced rapid disease progression (Fig. 6E). Although they comprised fewer than 8% of clinically early-stage patients (Fig. 6F), nearly 80% progressed, highlighting the high clinical need in this high-risk subgroup (Fig. 6G).

Taken together, our data support a clinical algorithm in which ~8% of patients at very high risk of systemic progression are identified early for DCC-derived model generation. For most of these patients, models will become available prior to progression, enabling drug screens to inform treatment decisions in time.

## Discussion

In this study, we explored if and how functional drug testing could be integrated into clinical practice before the manifestation of metastasis by generating DCC-derived models. Under the well-substantiated premise that systemically spread cancer cells differ molecularly and phenotypically from primary tumors (PT) (Werner-Klein et al, 2018; Guetter et al, 2025) and considering that melanoma PTs are often completely used for diagnostic purposes, we generated models from lymphatically spread cancer cells. These cells can be collected during routine staging procedures, such as SLN or NSLN biopsy. The generated models closely mirrored ex vivo isolated cells, were mainly available before progression, and enabled screening of a broad range of therapeutic options.

We explored several approaches to establish DCC-derived models, including xenograft generation and direct in vitro culture of disaggregated LN material, i.e., LN cell suspensions or pre-cultured DCC-derived spheres. For all approaches, we noted a link between success rate and the DCCD and LN type. We identified the DCCD as a valuable metric for selecting patients whose models are likely to be successfully generated and who are in greater need of additional therapies, as the DCCD is quantitatively associated with the risk of progression and death (Ulmer et al, 2014; Ulmer et al, 2018; Guetter et al, 2025). Using a threshold of DCCD ≥ 100, we predict that ~80% of patients will experience disease progression and therefore benefit from the availability of a patient-derived model enabling functional drug testing to identify additional therapeutic options for second-line treatment. Interestingly, ex vivo expansion of aberrant DCCs by sphere culture was observed at a similar DCCD in LN samples from lung cancer patients (Treitschke et al, 2023). This suggests that colony formation, as reflected by a DCCD ≥ 100, may define a critical step in the molecular, phenotypic, and functional evolution of solid cancers.

When establishing cell lines from patient-derived samples, the use of NSLNs resulted in a higher success rate. This may be rooted in the fact that involvement of NSLN reflects very high DCCD values. However, the observation that melanoma cells from NSLNs displayed higher numbers of CNAs and that LN type was an independent factor in multivariable analysis may indicate further biological differences between SLN and NSLN-derived melanoma cells relevant for CL generation. Since CLs could be established directly from patient samples without prior xenotransplantation in five cases (one SLN, four NSLNs, all with DCCD > 50,000), direct CL generation can be recommended for high-risk patients with very high DCCD counts. However, if the goal is to maximize the chance of expanding DCCs even from patients with low DCCD, using a PDX will be useful despite substantial resources in terms of cost, logistics, personnel, and regulatory effort.

Patient-derived preclinical models have been used to retrospectively analyze and understand therapy outcomes or predict susceptibilities to personalized therapies for individual patients in co-clinical trials (Gao et al, 2015). Although the drug responses observed in models often

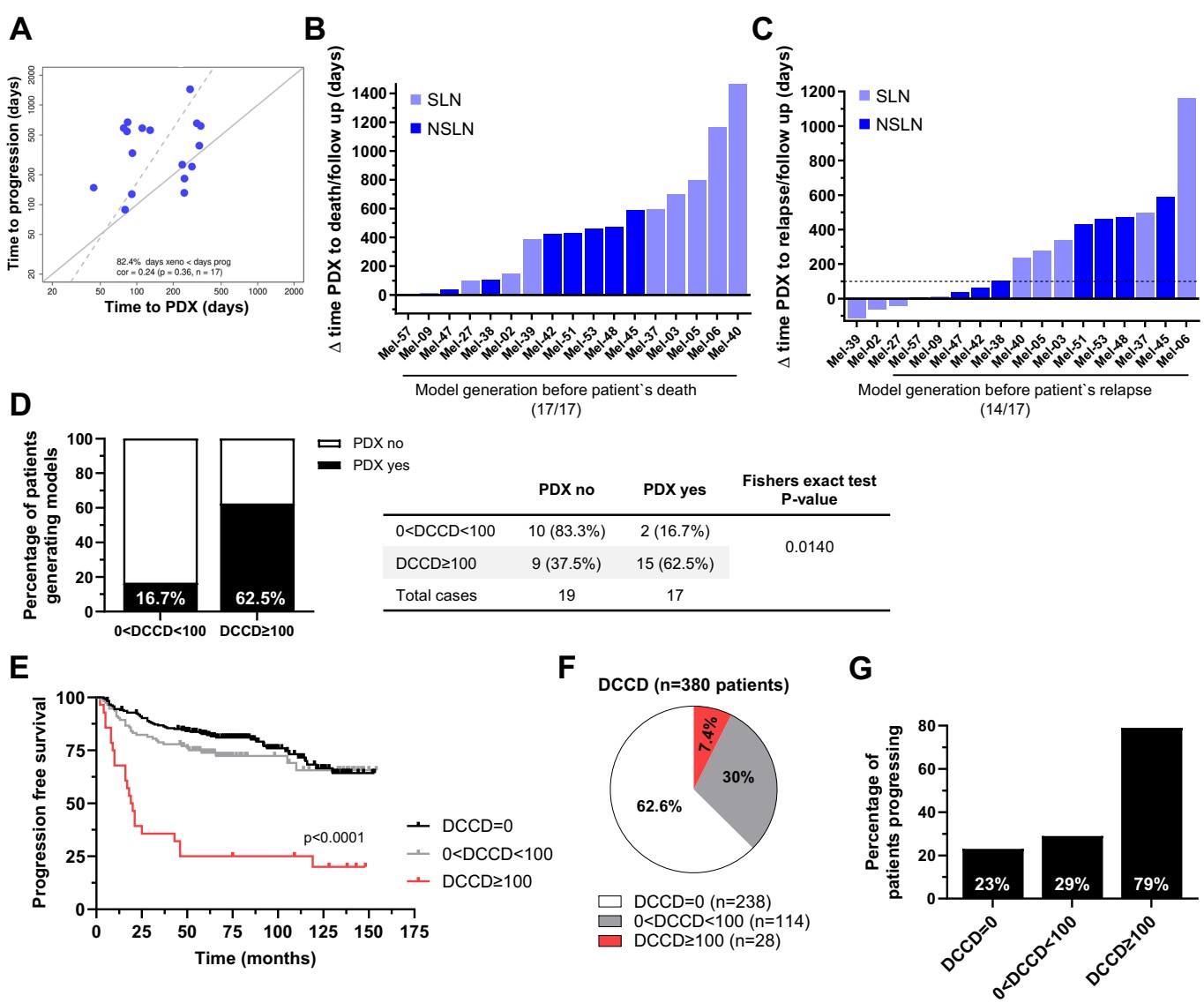

**Figure 6. Model generation and clinical implementation.**

(A) Correlation of the time needed for PDX generation (time from sample receipt to tumor harvest) and time to patient's disease progression (time to progression), $r = 0.24$. The analysis included the fastest available model for each patient. Each dot represents an individual patient. The gray solid and dashed lines represent the diagonal and the result of parametric linear regression, respectively. (B) Time between PDX generation and patient death (or last follow-up for surviving patients) in days. The fraction of patients from whom models could be generated before patient death is given (17/17). The analysis included the fastest available model for each patient. Light and dark blue bars indicate patients from whom SLN or NSLN were obtained, respectively. (C) Time between PDX generation and patient relapse. The fraction of patients from whom models could be generated before patients progressed is given (14/17). Bars indicate the "window of opportunity" in days to generate personalized molecular or functional information for treatment decisions. The dotted line indicates 100 days, the approximate time needed for in vitro expansion and drug-sensitivity testing results. Light and dark blue bars indicate patients from whom SLN or NSLN were obtained, respectively. (D) Percentage of patients categorized by their DCCD, who successfully engrafted as PDX (white = no PDX generated, black = PDX generated). $0 < DCCD < 100$ ($n = 12$ patients) and $DCCD \geq 100$ ($n = 24$ patients). P value according to Fisher's exact test, two-sided; *$P = 0.0140$. (E) Kaplan–Meier curves of progression-free survival of 380 patients stratified according to the DCCD of their SLN. DCCD = 0 ($n = 238$ patients), $0 < DCCD < 100$ ($n = 114$ patients), $DCCD \geq 100$ ($n = 28$ patients). P values were determined by log-rank test (DCCD = 0/ $0 < DCCD < 100$ vs. $DCCD \geq 100$, ****$P = 0.0001$). (F) Percentage of patients in each DCCD category. White indicates patients with DCCD = 0, ($n = 238$ patients), gray with $0 < DCCD < 100$ ($n = 114$ patients), and red with $DCCD \geq 100$ ($n = 28$ patients). (G) Percentage of patients experiencing progression in each DCCD group. DCCD categories as defined in (F). Source data are available online for this figure.

accurately recapitulate patient outcome (Izumchenko et al, 2017), co-clinical drug testing has not yet been implemented in routine personalized medicine programs. Molecular heterogeneity (McGrana-han and Swanton, 2017), particularly between PT and metastases, however, limits their direct application in early-stage disease. In contrast to PTs as the canonical cellular source for model generation, we hypothesize that DCCs that have already left the PT before surgery and started to form metastatic colonies are the most relevant target cells of adjuvant treatments. By assessing the genomic and histomorphological characteristics of DCC-derived models, we found a high

degree of similarity with ex vivo isolated, non-cultured DCCs or later arising metastases. Apparently, the selection of evolutionarily 'fit' DCC and their clonal expansion is associated with genome stabilization, which might explain the frequent failure to identify new druggable genetic targets in the case of metastatic progression. To better understand the DCC genotype and phenotype of Mel-DCC CLs at different passages and various culture conditions, in-depth analyses such as whole-exome/genome sequencing and transcriptome analyses will be required.

To experimentally test and determine whether functional resistance generated during treatment could be addressed in a timely manner, we exposed *BRAF*-mutant DCC-derived models to BRAF inhibition followed by resistance selection. While we were unable to identify any actionable mutation deemed causative of the emerging BRAF inhibitor resistance in Mel-DCC-11, functional drug testing revealed a number of promising therapeutic options, including HDAC-, microtubule-, topoisomerase-, or EGFR-inhibitors and drugs targeting the PI3K/Akt/mTOR signaling pathway, which have been described to overcome drug resistance in mutant BRAF melanoma cells (Dratkiewicz et al, 2019; Wang et al, 2021; Misek et al, 2022). The observed drug sensitivities are likely to be patient-specific and not broadly generalizable to all melanoma cases, given the unique genetic and epigenetic landscapes of each tumor and the patient's individual treatment history. However, our DCC models offer versatile tools for in-depth analyses, such as large-scale drug screenings, enabling the exploration of resistance mechanisms to guide the development of more effective therapeutic strategies. As more CLs are tested, we will learn to what extent the identified vulnerabilities and in vitro developed resistance mechanisms overlap with resistance in individual patients. In addition, the application of whole-exome or genome sequencing approaches will provide a more comprehensive understanding of how intrinsic or acquired drug resistance to targeted therapies is caused and can be effectively addressed.

Finally, we demonstrate that Mel-DCC/LN cocultures provide a feasible and informative in vitro platform for assessing T-cell activation and responsiveness to immune checkpoint ligands (ICLs) and their inhibition. Although autologous immune assays may more closely capture patient-specific immune dynamics, our findings support the usefulness of an allogeneic assay. The data show that the influence of ICLs expression on T-cell activation and their inhibition can be adequately tested even when allogeneic material is not available.

To use DCC-derived models for personalized treatment selection, it is essential that models and drug screen results become available prior to systemic progression. This was the case for about 80% of our models. In a clinical setting, disease progression of selected patients should be monitored continuously, for example by tracking CTC counts or ctDNA levels, which often precede imaging-determined relapse by several months (Pantel and Alix-Panabières, 2019; Cohen et al, 2023). Since early treatment at low tumor load is likely to improve outcome, the availability of alternative treatment options at this stage might prove useful. Importantly, it should be noted that patients in our study are not eligible for neoadjuvant therapy, which applies to those with clinically detected macroscopic, resectable disease according to current guidelines (Garbe et al, 2025). Instead, our study focused on earlier-stage patients without macroscopic, clinically detectable lymph node metastasis, for whom functional testing could become important for selecting adequate treatment options.

While the data presented here provide necessary information for designing future clinical study protocols, widespread application of the presented workflow will require replacing the use of xenograft generation with improved methods for in vitro expansion, which may become available soon. We anticipate that a thorough analysis of DCCs that either succeeded or failed in generating models will shed light on cell intrinsic and extrinsic factors determining expansion fate both in vivo and in vitro. Until then, a xenograft-based clinical trial could help determine whether functional drug screening can inform therapy decisions and improve clinical outcomes for the approximately 8% of melanoma patients who are in predictable need of additional therapies.

# Methods

**Reagents and tools table**

| Reagent/resource | Reference or source | Identifier or catalog number |
|---|---|---|
| **Experimental models** | | |
| Lymph nodes from melanoma patients | University Medical Center Regensburg | |
| NOD.Cg-*Prkdc*^scid^*IL2ry*^tmWjl/Sz^ | Jackson Laboratory | RRID:IMSR_JAX:005557 |
| Normal Human Epidermal Melanocytes | Lonza | Cat# CC-2586 |
| **Recombinant DNA** | | |
| **Antibodies** | | |
| Anti-Human Melanosome, HMB45 (monoclonal, mouse) | Dako | Cat# M0634 |
| Anti-Human Chondroitin Sulfate (monoclonal mouse) | BD Pharmingen/BD Biosciences | Cat# 554275 |
| Anti-Mouse IgG -AffiniPure F(ab')2 Fragment-Cy3 (polyclonal, goat) | Jackson ImmunoResearch | Cat# 115-166-071 |
| Anti-Human HLA Class I ABC-FITC (monoclonal mouse) | Acris | Cat# SM1222F; RRID:AB_1005115 |
| Anti-Human Melanoma (MCSP)-APC (monoclonal, mouse) | Miltenyi Biotec | Cat# 130-091-252; RRID:AB_871575 |
| Anti-Human Fibroblast REAfinity™-PE (monoclonal) | Miltenyi Biotec | Cat# 130-100-136; RRID:AB_2651739 |
| Anti-Human CD45-FITC (monoclonal, mouse) | BioLegend | Cat# 304005; RRID:AB_314393 |
| Anti-Human CD63-PE-Cy7 (monoclonal, mouse) | Thermo Fisher Scientific | Cat# 25-0639-41; RRID:AB_2573357 |
| Anti-Human CD146-eFluor™ 450 (monoclonal, mouse) | Thermo Fisher Scientific | Cat# 48-1469-41; RRID:AB_2574042 |
| Anti-Human CD166 (ALCAM)-PE (monoclonal, mouse) | Thermo Fisher Scientific | Cat# 12-1668-41; RRID:AB_2572586 |
| Anti-Human CD271 (NGFR)-FITC (monoclonal, mouse) | BioLegend | Cat# 345103; RRID:AB_1937226 |
| Anti-Human CD274 (PD-L1, B7-H1)-PE (monoclonal, mouse) | Thermo Fisher Scientific | Cat# 12-5983-42; RRID:AB_11042286 |

*EMBO Molecular Medicine*

Kathrin Weidele et al

| Reagent/resource | Reference or source | Identifier or catalog number |
|---|---|---|
| Anti-Mouse CD29-PE (monoclonal, hamster) | Miltenyi Biotec | Cat# 130-102-602; RRID:AB_2660694 |
| Anti-Mouse CD45-eFluor™ 450 (monoclonal, rat) | Thermo Fisher Scientific | Cat# 48-0451-82; RRID:AB_1518806 |
| Anti-Human Ki-67-Alexa Fluor® 488 (monoclonal, mouse) | BioLegend | Cat# 350507, RRID:AB_10900418 |
| Anti-Human CD14-FITC (monoclonal, mouse) | Thermo Fisher Scientific | Cat# 11-0149-42, RRID:AB_10597597 |
| Anti-Human CD3-PerCP (monoclonal, mouse) | BioLegend | Cat# 344813, RRID:AB_10641841 |
| Anti-Human CD19-PE (monoclonal, mouse) | Thermo Fisher Scientific | Cat# 12-0199-42, RRID:AB_1834376 |
| Anti-Human CD197 (CCR7)-PE-Cy7 (monoclonal, mouse) | BioLegend | Cat# 353225, RRID:AB_11125576 |
| Anti-Human CD56 (NCAM)-APC (monoclonal, mouse) | BioLegend | Cat# 318309, RRID:AB_604098 |
| Anti-Human CD45RA Alexa Fluor® 700 (monoclonal, mouse) | BioLegend | Cat# 304119, RRID:AB_493762 |
| Anti-Human CD4-Brilliant Violet 421™ (monoclonal, mouse) | BioLegend | Cat# 317434, RRID:AB_2562134 |
| Anti-Human CD8a-Brilliant Violet 510™ (monoclonal, mouse) | BioLegend | Cat# 300933, RRID:AB_2814114 |
| Anti-Human CD25-PE, clone BC96 (monoclonal, mouse) | BioLegend | Cat# 302606, RRID:AB_314276 |
| Anti-Human CD279 (PD-1)-PE-Cy7 (monoclonal, mouse) | BioLegend | Cat# 329918, RRID:AB_2159324 |
| Anti-Human CD4-APC, REAfinity™ (monoclonal) | Miltenyi Biotec | Cat# 130-113-222, RRID:AB_2726033 |
| Anti-Human CD366 (TIM-3)-Brilliant Violet 421™ (monoclonal, mouse) | BioLegend | Cat# 345007, RRID:AB_10900073 |
| Anti-Human CD274 (PD-L1)-PE (monoclonal, mouse) | BioLegend | Cat# 329705, RRID:AB_940366 |

**Oligonucleotides and other sequence-based reagents**

Not applicable

**Chemicals, enzymes, and other reagents**

| | | |
|---|---|---|
| Diagnostic microscopic slides | Thermo Scientific | Cat# ER-203B-CE24 |
| DMEM/F12 | PAN-Biotech | Cat# 41450 |
| RPMI 1640 | PAN-Biotech | Cat# P04-17500 |
| MGM-4 melanocyte growth media-4 BulletKit™ | Lonza | Cat# CC-3249 |
| Poly-HEMA, Poly(2-hydroxyethylmethacrylat) | Sigma-Aldrich | Cat# P3932 |
| B-27 Supplement | Gibco | Cat# 17504044 |
| Recombinant Human CXCL1/GRO alpha Protein | R&D Systems | Cat# 275-GR-010 |

| Reagent/resource | Reference or source | Identifier or catalog number |
|---|---|---|
| Hyper-IL-6 | Provided by S. Rose-John | |
| Mouse Embryonic Fibroblast (MEF) Conditioned Media | R&D Systems | Cat# AR005 |
| Methylcellulose | Sigma-Aldrich | Cat# M0512 |
| Matrigel® Basement Membrane Matrix | Corning | Cat# 354248 |
| Ampli1™ WGA Kit | Menarini | Cat# WG001R |
| Human AB Serum | BioIVT | Cat# HUMANABSRMP-HI-1 |
| DNeasy Blood & Tissue Kit | Qiagen | Cat# 69504 |
| SureTag DNA Labeling Kit | Agilent | Cat# 5190-3400 |
| Oligonucleotide-based SurePrint G3 Human CGH 4x180K microarray slides) | Agilent | Design code: 022060 |
| GenePrint® 10 System | Promega | Cat# B9510 |
| Vemurafenib (PLX4032) | Selleckchem | Cat# S1267 |
| Binimetinib (MEK162) | Selleckchem | Cat# S7007 |
| Pembrolizumab (anti-PD-1) | Selleckchem | Cat# A2005 |
| Nivolumab (anti-PD-1) | Selleckchem | Cat# A2002 |
| Ipilimumab (anti-CTLA-4) | Selleckchem | Cat# A2001 |
| Anti-Cancer Approved Drug Library (315 compounds) | TargetMol | Cat# L2110 |
| CellTiter-Blue® Cell Viability Assay | Promega | Cat# G8081 |
| ATPlite 1step Luminescence Assay System | PerkinElmer | Cat# 6016739 |
| Zombie NIR™ Fixable Viability Kit | BioLegend | Cat# 423105 |
| FluoroFix™ Buffer | BioLegend | Cat# 422101 |
| Intracellular Staining Permeabilization Wash Buffer (10X) | BioLegend | Cat# 421002 |
| Human IL-2 IS | Miltenyi Biotec | Cat# 130-097-743 |
| Dynabeads™ Human T-Activator CD3/CD28 | Thermo Fisher Scientific | Cat# 11161D |

**Software**

| | | |
|---|---|---|
| Agilent Feature Extraction 10.7 | Agilent | |
| Agilent Genomic Workbench 6.5 | Agilent | |
| Biorender | https://www.biorender.com | |
| FlowJo 10.10.0 | BD | |
| GraphPad Prism 10 | https://www.graphpad.com | |
| Kaluza Analysis 2.2 | Beckman Coulter | |
| R 4.4.2, R 5.4.1 | https://www.r-project.org | |

**Other**

## Ethics and patients

We used data from 57 melanoma patients from Regensburg who underwent tumor resection, sentinel lymph node biopsies, or lymph adenectomies. The study was approved by the University of Regensburg Ethics Committee (ethics vote 07-079) and conducted in accordance with the WMA Declaration of Helsinki and the Department of Health and Human Services Belmont Report. All patients provided written informed consent.

## Patient sample preparation and DCC detection

Quantitative immunocytology of unfixed LN tissues was performed as described before (Ulmer et al, 2014; Werner-Klein et al, 2018). In brief, bisected LNs were mechanically disaggregated into a single-cell suspension and enriched via 60% Percoll gradient centrifugation (Ulmer et al, 2005). For DCC detection and quantification, $10^6$ cells were transferred onto adhesion slides (Thermo Scientific) and stained for gp100 (HMB45, Dako, dilution 1:100). The number of positive cells per million lymphocytes, defined as DCCD (disseminated cancer cell density), was determined and documented. Gp100-positive cells were manually picked from the adhesion slides using a micromanipulator (Eppendorf PatchMan NP2) and subjected to whole-genome amplification and array comparative genomic hybridization procedure as described below.

For the isolation of viable DCCs, cell suspensions were stained with an anti-human MCSP antibody (BD Pharmingen, dilution 1:50) and detected by indirect immunofluorescence (goat anti-mouse-Cy3, Jackson ImmunoResearch, dilution 1:100). MCSP$^+$ cells were isolated using a micromanipulator and transplanted as described below.

## In vitro sphere cell culture

Disaggregated LN cells were plated in 6-cm poly-HEMA-coated (12 mg/ml, Sigma-Aldrich) cell culture plates at a density of 200,000 viable cells/ml. Cells were grown under hypoxic conditions (7% $O_2$) at 37 °C in 5 ml serum-free DMEM/Ham's F12 basal medium (PAN-Biotech), supplemented with 0.5% penicillin/streptomycin (both PAN-Biotech), 0.5% BSA (VWR), 10 µg/ml insulin (Sigma-Aldrich), 10 nM HEPES (Sigma-Aldrich), 1× B27 (Life Technology), 10 ng/ml EGF (Sigma-Aldrich) and 10 ng/ml bFGF (Sigma-Aldrich), 4 µg/ml heparin (Sigma-Aldrich), 5 ng/ml GRO-α (R&D Systems), 20 ng/ml HIL-6 (kindly provided by S.Rose-John), mouse embryonic fibroblast conditioned medium (1:5, R&D Systems) and 0.2% methylcellulose (Sigma-Aldrich). Cultures were monitored and complemented with 500 µl fresh medium once a week. Median time from sphere generation to xenotransplantation was 25 days (range, 8–43 days).

## Xenotransplantations

Animals were purchased from the Jackson Laboratory and maintained under specific pathogen-free conditions, with acidified water and food ad libitum in the research animal facilities of the University of Regensburg, Germany. All studies were conducted in compliance with European guidelines for the care and use of laboratory animals and were approved by the Institutional Animal Care and Use Committees (IACUC) of the Universitätsklinikum Regensburg and Regierung Unterfranken.

Spheres or MCSP$^+$ cells were transferred in a poly-HEMA-coated microwell (volume 10–15 µl, Terasaki Plates) and subsequently transplanted in a final volume of 30 µl and 25% high-concentration matrigel (Corning) as published before (Quintana et al, 2008; Werner-Klein et al, 2018). Depending on availability, between seven to 200,000 cells (median = 37 cells) or spheres in the range of three to 107 (median = 25 spheres) were transplanted. Samples were injected subcutaneously into 6–10-week-old NSG mice (NOD.Cg-$Prkdc^{scid}IL2ry^{tmWjl/Sz}$). Injection sites were monitored every week for tumor growth. When the maximal tumor burden was reached or mice showed signs of illness, animals were euthanized and tumors were harvested.

## Generation of patient-derived DCC cell lines and cell culture

For the generation of patient-derived in vitro models, several approaches were pursued. CL were either generated from PDX tumors by enzymatic dissociation (Hosseini et al, 2016) and cultivation under adherent culture conditions (approach I) or without PDX generation by culturing dissociated LN suspensions or pre-cultured DCC-derived spheres under plastic-adherent 2D culture conditions (approach II). Cell suspensions were analyzed before and during cultivation for human origin (HLA class I ABC) and human melanoma-specific surface markers (Reagents and Tools Table, antibody dilutions 1:100) using flow cytometry (Gallios, Beckman Coulter). Cultures were incubated in RPMI 1640 with 10% FBS premium, 100 U/ml penicillin, 0.1 mg/ml streptomycin (all from PAN-Biotech) and 1× GlutaMAX Supplement (Gibco) at 37 °C and 5% $CO_2$. In addition to testing for the presence of human melanoma-specific surface markers, cell cultures were assessed by flow cytometry for the absence of non-melanoma human cells (CD45 and fibroblasts) and mouse cells (CD29 and CD45) at least every five passages, up to passage 20. All CLs grew as adherent cultures and could be cryopreserved, thawed, and regrown.

Drug-resistant sublines (Mel-DCC-03-R, Mel-DCC-05-R, Mel-DCC-10-R, Mel-DCC-11-R) were established from the drug-sensitive parental CL by continuous exposure to increasing concentrations of Vemurafenib (PLX4032, Selleckchem) or Binimetinib (MEK162, Selleckchem).

Normal Human Epidermal Melanocytes (NHEM) were purchased and cultured in MGM-4 melanocyte growth media-4 BulletKit™ from Lonza. NHEM were transduced with pLV-hTERT-IRES-hygro lentiviral particles (Addgene) to generate immortalized melanocytes (NHEM-hTERT).

## Coculture of Mel-DCC cell lines and LN cells

Frozen patient-derived lymph node suspension cells (LN-01, LN-02, and LN-03, all DCC-negative) were rapidly thawed, washed, and rested overnight at a density of $1 \times 10^7$ cells/ml in T-cell medium (RPMI 1640 supplemented with 10% human AB serum (BioIVT) and 1% penicillin–streptomycin) at 37 °C and 5% $CO_2$. The following day, Mel-DCC cells were seeded at $2 \times 10^4$ cells per well in 100 µl of T-cell medium in U-bottom 96-well plates (Corning). Subsequently, $2 \times 10^5$ LN cells were added in 100 µl of

the same medium and mixed gently by pipetting, resulting in an LN:Mel-DCC ratio of 10:1. Cultures were supplemented with recombinant human IL-2 (100 IU/ml, Miltenyi Biotec). Control conditions included medium alone (control), IL-2 only (resting), or Dynabeads™ Human T-Activator CD3/CD28 (Thermo Fisher Scientific) at an LN:bead ratio of 1:1. Immune checkpoint inhibitors were added at a final concentration of 10 μg/ml. Cocultures were maintained for 7 days and subsequently analyzed by flow cytometry ("Reagents and Tools Table", antibody dilutions 1:100).

## Immunohistochemistry (IHC)

IHC was performed on formalin-fixed, paraffin-embedded 4-μm tissue sections using an antibody to human melanosome (HMB45, anti-human gp100 melanosome, Dako, dilution 1:100) according to routine laboratory staining protocols and evaluated by an experienced pathologist using light microscopy (Leitz DMRBE microscope). Gp100 expression in primary tumors, PDX, and metastases was assessed semiquantitatively by means of the commonly used H-score, where staining intensity ranging from 0 (negative) to 3 (strong) is multiplied by the percentage of positively stained tumor cells and added for the final score.

## Whole-genome amplification (WGA)

DNA of isolated single cells and spheres was amplified using the Ampli1™ WGA Kit (Menarini, Silicon Biosystems) according to the manufacturer´s instructions.

## Analysis of copy number alterations (CNA)

For array comparative genomic hybridization (aCGH), genomic DNA was extracted using the DNeasy Blood & Tissue Kit (Qiagen). Genomic labeling was performed using the Agilent SureTag DNA Labeling Kit. Genome-wide copy number analysis was conducted using the array comparative genomic hybridization protocol as previously published (Czyż et al, 2014). WGA-amplified samples and corresponding, gender-matched reference DNA (a mixture of four WGA products, each generated from a single peripheral blood lymphocyte of a healthy donor, mixed in even proportion), were labeled using a PCR-based labeling approach and subsequently hybridized on CGH microarray slides. Slides were scanned using an Agilent Microarray Scanner Type C, and images were processed with Agilent Genomic Feature Extraction Software. Resulting files were imported and analyzed with Agilent Genomic Workbench Software. Aberrant regions were recognized using ADM-2 algorithm with threshold set to 6.5. To set the most common ploidy to log2 = 0 centralization algorithm was set to a threshold of 6.0 and bin size of 10. To avoid false positive calls, an aberration filter was applied defining the following criteria for aberrant regions: (i) minimum absolute average log2 of 0.25 and (ii) minimum number of probes in an aberrant interval of 50.

Detection of CNAs using low-pass whole-genome sequencing was done for samples Mel-DCC-10a, -10b, and Mel-DCC-11 as published before (Treitschke et al, 2023; Guetter et al, 2025).

ArrayCGH raw data derived from Agilent Workbench were median-centered and scaled by median absolute deviation. Subsequently, sample profiles were processed as described in (Hosseini et al, 2016). The QDNAseq-derived low-pass segmentation data

were filtered for missing values and integrated with the segmented aCGH data by coarse-graining (median-summarizing) the array data. The dendrogram was obtained by hierarchical clustering of samples using Euclidean distance and complete linkage. Calculations were performed using R, v4.5.1, and the R-packages RColorBrewer, v1.1.3, and ComplexHeatmap, v2.24.1.

## DNA fingerprinting

Patient-origin was authenticated using short tandem repeat analysis (GenePrint10, Promega). Due to the WGA of DCC and sphere samples, which includes restriction digestion by Mse I prior to STR analysis, in some cases only STR loci TH01, D21S11, D5S818, D13S317, D16S538, and vWA were accessible for analysis. Amplified fragments were detected using 3130-Avant Genetic Analyzer (Applied Biosystems). Fragment sizes were determined and evaluated using GeneMapper® Software 4.1. STR profiles of PDX samples and CLs were compared to the profile of the corresponding DCC. All analyzed patient samples matched each other (Fig. EV1A). The obtained STR profiles were additionally analyzed using a public database (www.cellosaurus.org/str-search/). This comparison showed no match with other cell lines. Patient samples are therefore revealed as unique and not cross-contaminated or misidentified with cells of other origin.

Hierarchical sample clustering was done according to STR features using Euclidean distance and complete linkage. Calculations were performed using R, v4.5.1, and the R-packages DNAcopy, v1.82.0, RColorBrewer, v1.1.3, and ComplexHeatmap, v2.24.1.

## Mutation analysis of BRAF and NRAS

Mutations in NRAS and BRAF genes were detected using Sanger sequencing after locus-specific end-point PCR conducted on WGA samples, as published before (Werner-Klein et al, 2018).

## Targeted panel sequencing

Genomic DNA was extracted using the DNeasy Blood & Tissue Kit (Qiagen) and subsequently sheared to an average fragment size of 180-220 bp using the Covaris S220X Ultrasonicator (Covaris). Sequencing libraries were prepared with the Kapa HyperPrep Kit (Roche), following the manufacturer's instructions for the KAPA EZ HyperCap Workflow, version 2.3. Libraries were then pooled for multiplexed hybridization capture using the MSK-IMPACT (Memorial Sloan Kettering-Integrated Mutation Profiling of Actionable Cancer Targets) gene panel, version 2 (410 genes; NimbleGen), utilizing biotinylated DNA probes. Captured libraries were sequenced on a NovaSeq 6000 system (Illumina) to generate 150 bp paired-end reads.

Paired-end reads from the DCC-CLs and matching normal peripheral blood lymphocytes were processed using the in-house sequencing workflow v0.2.0 of HIENA v3.2.0. Adapter sequences and poor quality bases were trimmed using BBDuk v38.84 (Joint Genome Institute-JGI DataScience, 2019), and BioBloom tools v2.0.13 (Chu et al, 2014) were used to remove microbial/fungal DNA contaminants plus any other general lab contaminants. BWA-MEM v0.7.17 (Li, 2013) was used to map the trimmed and decontaminated reads to the Human reference genome GRCh38 GCA_000001405.15. Further, marking of duplicate reads and base score quality recalibration were

done using Picard v2.21.8 (Broad Institute, 2019) and GATK v4.2.6.1 (van der Auwera et al, 2013), respectively. Finally, the bam output from this step was used for variant calling. MuTect2 in GATK 4.2.6.1, Freebayes 1.3.6 (Garrison and Marth, 2012), and Strelka 2.9.10 (Kim et al, 2018) were used to call variants in the tumor-matched normal bam files with default parameters. BCFtools merge was used to obtain one concatenated VCF (Variant Caller Format) output. Finally, the variants were annotated using Annovar (downloaded on 2020-06-08) (Wang et al, 2010), Variant Effect Predictor (VEP) 105 (McLaren et al, 2016) and SnpEff 5.0 (Cingolani et al, 2012). Standard filters including alternate allele depth AD in tumor samples of more than 5, allele frequency AF (read depth in DCC-CL sample/read depth in normal sample) more than 0.1 and total depth DP across both samples of more than 10 for filtering the mutation calls were applied. Mutations were selected based on their pathogenicity and whether they were potential mutations in known oncogenic genes.

## Drug-sensitivity testing

For drug-sensitivity assays, 1000 cells per well were seeded in 96-well plates (VWR) in growth medium for 24 h. Then, cells were treated in triplicates with a single dose Vemurafenib with doses ranging from 0.025 μM to 5 μM, or Binimetinib with doses ranging from 0.001 μM to 1 μM for 120 h. In total, 500 μM hydrogen peroxide treatment was used as a positive control. CellTiter-Blue® Cell Viability Assay reagent (Promega) was added according to the manufacturer's protocol. Fluorescence was recorded with a fluorescence reader (TECAN GENios).

To identify drug targets involved in overcoming drug resistance, we used a drug library comprising 315 anti-cancer approved drugs (TargetMol) and screened NHEM-hTERT, Mel-DCC-11, and Mel-DCC-11-R (in the presence and absence of 8 μM Vemurafenib) at one concentration (1 μM) in duplicates. In all, 2500 NHEM-hTERT cells/well or 1500 Mel-DCC-11/Mel-DCC-11-R cell/well (40 μl) were added to 384-well plates (Greiner) pre-spotted with the library by Echo acoustic dispensing (Beckman Coulter) and incubated for 120 h. Cell viability was measured after addition of the ATPlite 1step Luminescence Assay System (30 μl/well) using the EnVision plate reader (both PerkinElmer). Values were normalized to DMSO controls and expressed as % viability. Information on drug targets, mechanisms of action, and pathways were extracted from the library information provided by the vendor and harmonized using DrugBank, ChEMBL, and the Clue database.

## Statistical analysis

Statistical analysis and data presentation were performed with GraphPad Prism Software 10 using the two-tailed Mann–Whitney test, one-way ANOVA for paired samples or Fisher's exact test to determine associations between two categorical variables. For correlation analysis, we used Spearman correlation and for survival analysis log-rank test for comparing survival curves. For univariable and multivariable analysis, R version 4.4.2 was employed. The $IC_{50}$ values and dose–response curves were calculated and plotted using the GraphPad Prism software based on a sigmoidal dose–response equation. Data representations and statistical details can be found in the description of the figure legends. All available samples that met predefined quality criteria were included, and exclusions were limited to technical artifacts. No blinding was performed.

### The paper explained

**Problem**

Most patients with melanoma, when diagnosed before manifestation of distant metastasis, are cured by timely surgery. However, a subset of patients will relapse and develop systemic cancer spread—a condition that is difficult to treat and very often fatal. Therefore, we tested whether cancer cells hidden in LNs can be expanded and used for functional drug testing for therapy selection.

**Results**

We used LN samples resected from the proximity of the primary melanoma to search for occult metastatic melanoma cells and established a protocol for cell expansion. We compared direct culture and expansion in mice. The major determinant for successful cell line generation was the number of melanoma cells per million LN cells. Since patients with high numbers are also at the highest risk for disease recurrence, the generation of cell lines of their melanoma for drug screening is most important. We showed exemplarily that several hundred drugs, including immunotherapies, can be tested in vitro. In all cases, models became available before the death of the patients.

**Impact**

The workflow may be clinically explored in 8% of early-stage melanoma patients with the highest relapse risk. Timely drug screening during the clinical latency period may provide information about effective treatment options to be administered at the earliest evidence of recurrent disease.

## Graphics

Figure 1A and the visual abstract were created with BioRender.com.

## Data availability

The panel sequencing data from this publication have been deposited to the EGA server under the accession number EGAS50000001225. Access to patient-derived material and raw sequencing data is restricted due to patient consent and compliance with the General Data Protection Regulation (GDPR).

The source data of this paper are collected in the following database record: biostudies:S-SCDT-10_1038-S44321-025-00339-8.

## Peer review information

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

## Acknowledgements

We thank Sybille Vorbeck, Siegfried Rein, Irene Nebeja, Theresa Kettel, Judith Proske, and Thomas Schamberger for excellent technical assistance. We also thank Philipp Renner for assistance in the recruitment of patient samples. We would like to express gratitude to the patients who participated in the study. This work was supported by the BayGene programme of the Bavarian State Ministry of Sciences, Research and the Arts, project 109753, by the Deutsche Krebshilfe (70113472 and 70112463), Wilhelm Sander Stiftung (2020.129.1), grants of the Deutsche Forschungsgemeinschaft (DFG, German Research Foundation: KL 1233/2-1, SFB-TRR 305-Z02), and the Bavarian Ministry of Economic Affairs, Energy and Technology (AZ 20-3410.1-1-1 and 20-3410.1-1-2), all to CAK; and grants to M.W.K. (WE4632/4-1, SFB TRR305-B09) and CW (SFB-TRR 305-B13).

## Author contributions

**Kathrin Weidele**: Data curation; Formal analysis; Supervision; Investigation; Visualization; Methodology; Writing—original draft; Writing—review and editing. **Christian Werno**: Conceptualization; Formal analysis; Supervision; Funding acquisition; Investigation; Visualization; Methodology; Writing—original draft; Project administration; Writing—review and editing. **Steffi Treitschke**: Data curation; Formal analysis; Supervision; Investigation; Visualization; Methodology; Writing—original draft; Writing—review and editing. **Catherine Botteron**: Data curation; Investigation; Methodology. **Martin Hoffmann**: Data curation; Software; Formal analysis; Investigation; Visualization. **Sebastian Scheitler**: Investigation. **Lukas Wöhrl**: Formal analysis; Investigation; Visualization; Writing—original draft. **Zbigniew Czyz**: Data curation; Investigation; Methodology. **Giancarlo Feliciello**: Investigation. **Florian Weber**: Resources; Investigation. **Adithi Ravikumar Varadarajan**: Data curation; Software; Formal analysis; Visualization. **Jens Warfsmann**: Data curation; Software; Formal analysis. **Silvia Materna-Reichelt**: Investigation. **Marie Katzer**: Resources; Investigation. **Laura Schreieder**: Resources; Investigation. **Parvaneh Mohammadi**: Investigation. **Hedayatollah Hosseini**: Investigation. **Kamran Honarnejad**: Supervision; Methodology. **Sebastian Haferkamp**: Resources; Investigation. **Melanie Werner-Klein**: Conceptualization; Supervision; Funding acquisition; Investigation; Methodology; Writing—original draft; Writing—review and editing. **Christoph A Klein**: Conceptualization; Resources; Supervision; Funding acquisition; Methodology; Writing—original draft; Project administration; Writing—review and editing.

Source data underlying figure panels in this paper may have individual authorship assigned. Where available, figure panel/source data authorship is listed in the following database record: biostudies:S-SCDT-10_1038-S44321-025-00339-8.

## Funding

## Disclosure and competing interests statement

The authors declare no competing interests.

# Expanded View Figures

## A

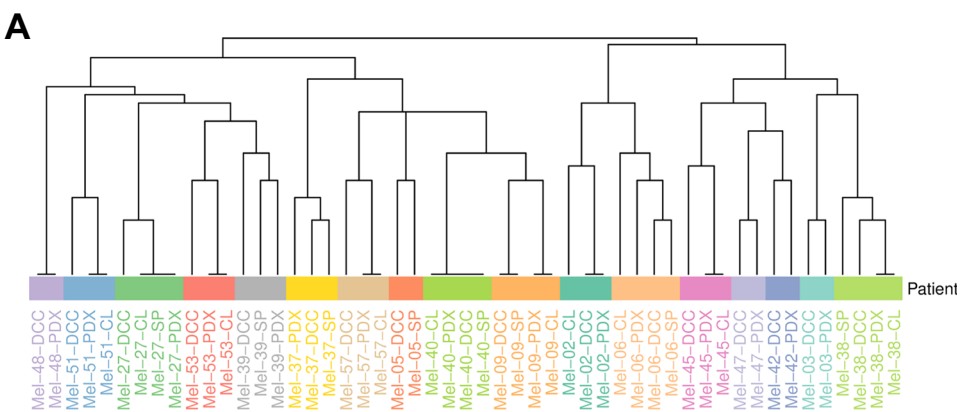

## B

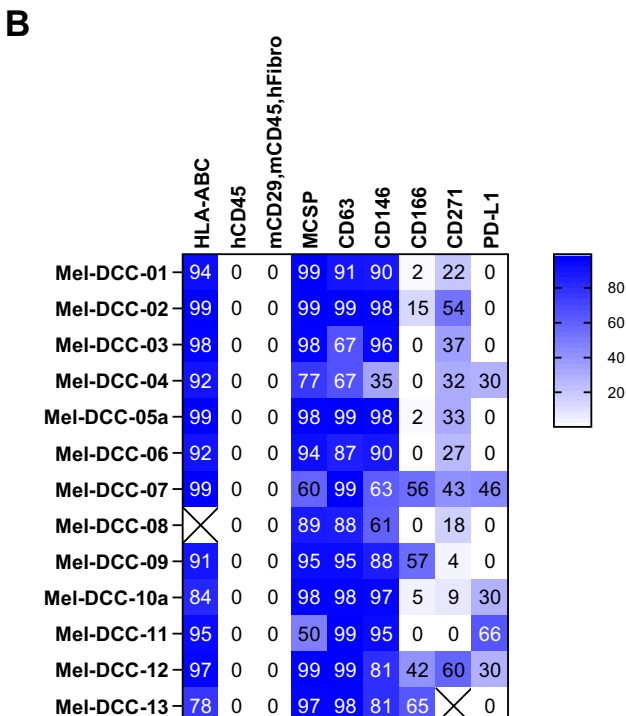

**Figure EV1. Sample and Mel-DCC CL characterization.**

(A) Cluster analysis of LN-DCCs, ex vivo cultured spheres (SP), PDX and CL models based on STR data. Hierarchical sample clustering according to 35 STR features using Euclidean distance and complete linkage. Analysis of Mel-05-PDX failed owing to sample quality issues. (B) Flow cytometry analysis of melanoma marker (MCSP, CD63, CD146, CD166, CD271) and immune checkpoint ligand (PD-L1) expression on Mel-DCC CLs. The human origin was verified using an anti-human HLA-ABC antibody. To exclude contaminations and ensure purity, CLs were tested for non-melanoma human cells (CD45, fibroblasts) and mouse cells (CD29, CD45). The scores (color scale) indicate the percentage of marker-positive cells compared to isotype controls.

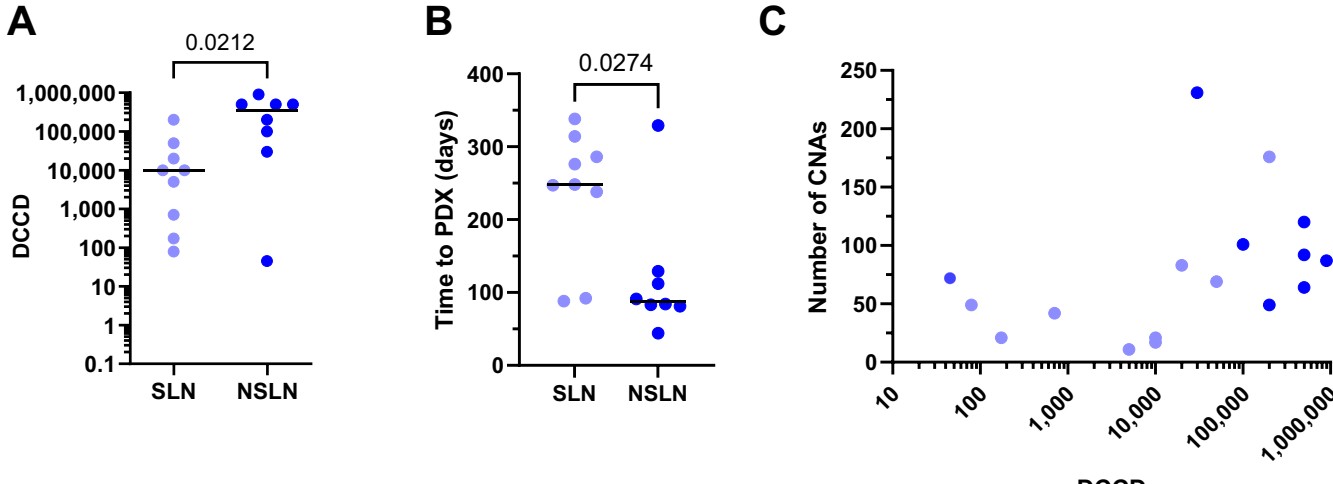

**Figure EV2. Association of DCCD and LN type, time to PDX generation and number of aberration calls in PDX.**

(A) Comparison of DCCD of SLNs and NSLNs that successfully formed PDXs by approach I (SLNs: $n = 9$ patients, NSLNs: $n = 8$ patients). Each dot represents an individual patient, the horizontal line indicates the median. *P* value according to an unpaired, nonparametric Mann–Whitney test; *$P = 0.0212$. Light and dark blue indicate SLNs and NSLNs, respectively. (B) Comparison of the time needed for PDX generation (time from sample receipt to tumor harvest) for SLNs and NSLNs. The analysis included the fastest available model for each patient. Each dot represents an individual patient, the horizontal line indicates the median. *P* value according to an unpaired, nonparametric Mann–Whitney test; ns, $P = 0.0274$). Light and dark blue indicate SLNs and NSLNs, respectively. (C) Correlation of DCCD and number of CNAs. Spearman *r* correlation, $r = 0.5603$; *$P = 0.0210$. Each plot represents an patient, the horizontal line indicates the median. Light and dark blue indicate SLNs and NSLNs, respectively.

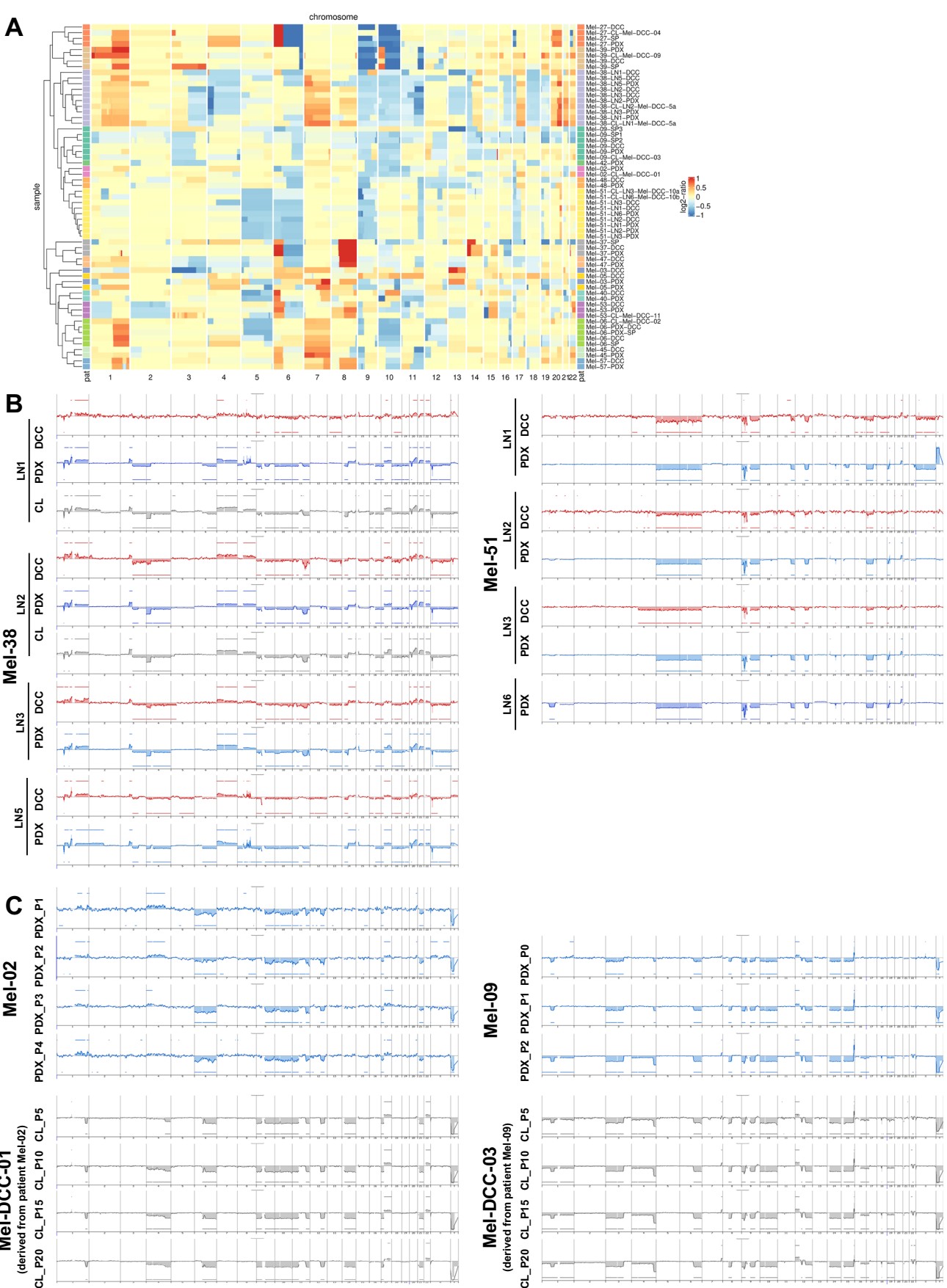

◄ **Figure EV3. Genome-wide CNA profiles of DCC-derived xenograft models (PDX), matched pairs of DCCs, DCC-derived spheres (SP) or in vitro-generated cell lines (CL).**

(A) Clustering of $\log_2$-fluorescence ratios of 4407 chromosomal bins using Euclidean distance and complete linkage. Negative $\log_2$ ratios indicate genomic losses, positive $\log_2$ ratios genomic gains relative to the median copy number. (B) CNA profiles of 2 patients, from which multiple LNs and corresponding models were obtained. (C) CNA profiles of PDX and CL models derived from patient Mel-02 and Mel-09. Serial in vivo/in vitro passages are displayed, respectively.

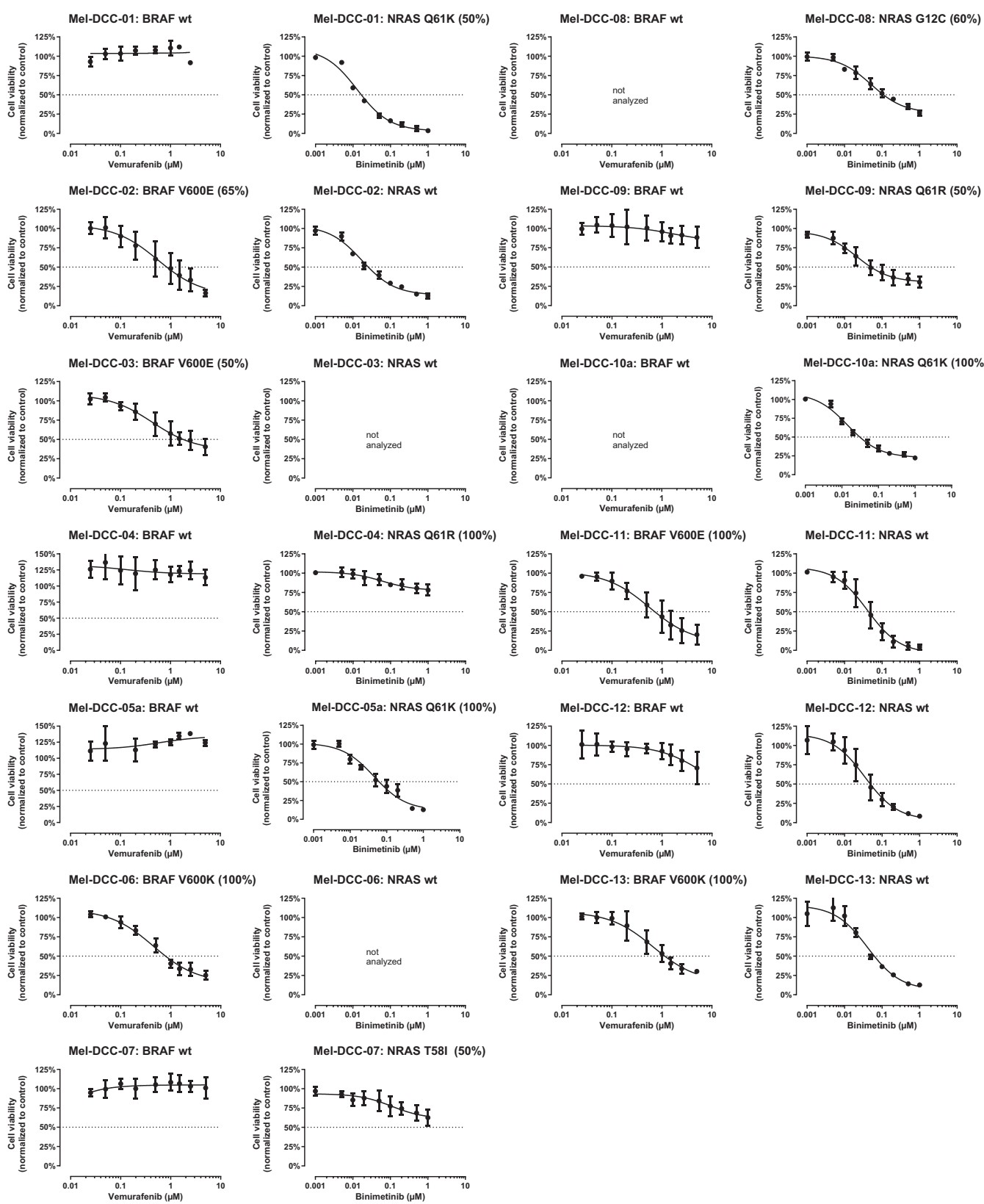

◀ **Figure EV4.  Dose response to targeted therapies in Mel-DCC CLs.**

Mel-DCC CLs were incubated for 5 days with doses ranging from 0.025 μM to 5 μM for Vemurafenib and from 0.001 μM to 1 μM for Binimetinib. Cell viability was monitored with CellTiter-Blue®. Dots indicate mean values ± SD of three biological replicates. Variant allele frequency is indicated in parentheses. Gray dotted lines indicate the $IC_{50}$ values (also given in Table 2).

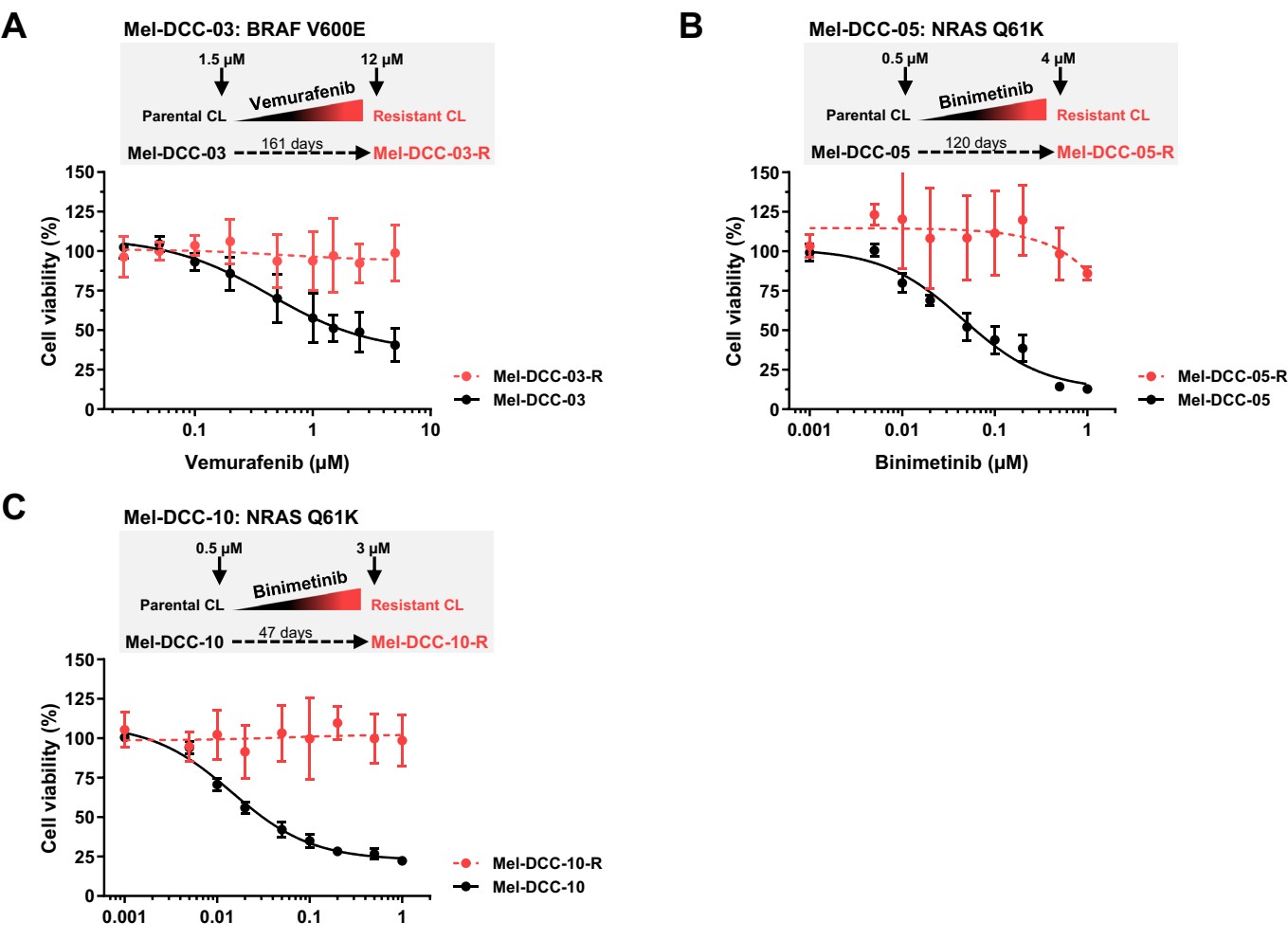

**Figure EV5. Generation of resistant Mel-DCC CLs.**

(A) Generation of a Vemurafenib-resistant BRAF V600E-mutated line (Mel-DCC-03-R). The resistant line (red) was generated through stepwise exposure of the parental line (Mel-DCC-03, black) to increasing concentrations of Vemurafenib over the indicated timeframe. Each dot indicates the mean ± SD of biological replicates (Mel-DCC-03, $n = 4$; Mel-DCC-03-R, $n = 5$). (B, C) Generation of Binimetinib-resistant NRAS-mutated melanoma cell lines. Sensitivity of Mel-DCC-05 vs. Mel-DCC-05-R (B) and Mel-DCC-10 vs. Mel-DCC-10-R (C) to Binimetinib are shown. Resistant lines (red) were generated through stepwise exposure of parental lines (black) to increasing concentrations of Binimetinib over the indicated timeframe. Each dot indicates the mean ± SD of biological replicates (Mel-DCC-05 and -10, $n = 4$; Mel-DCC-05-R and -10-R, $n = 5$).

