## [Peer Review File · EMBO Molecular Medicine]

Micrometastasis-derived models enable drug testing for early-stage, high-risk melanoma patients

Kathrin Weidele, Christian Werno, Steffi Treitschke, Catherine Botteron, Martin Hoffmann, Sebastian Scheitler, Lukas Wöhr, Zbigniew Czyz, Giancarlo Feliciello, Florian Weber, Adithi Ravikumar-Varadarajan, Jens Warfsmann, Silvia Materna-Reichelt, Marie Katzer, Laura Schreieder, Parvaneh Mohammadi, Hedayatollah Hosseini, Kamran Honarnejad, Sebastian Haferkamp, Melanie Werner-Klein, and Christoph Klein

Corresponding authors: Christoph Klein (christoph.klein@ukr.de) , Melanie Werner-Klein (melanie.werner-klein@ukr.de)

Review Timeline:

Submission Date:	9th May 25
Editorial Decision:	2nd Jun 25
Revision Received:	29th Sep 25
Editorial Decision:	21st Oct 25
Revision Received:	24th Oct 25
Accepted:	30th Oct 25

Editor: Zeljko Durdevic

Transaction Report:

2nd Jun 2025

Dear Dr. Klein,

Thank you for the submission of your manuscript to EMBO Molecular Medicine. We have now received feedback from the three reviewers who agreed to evaluate your manuscript. All three referees recognize interest of the study but also raise serious concerns that should be addressed in a major revision. If you would like to discuss further the points raised by the referees, I am available to do so via email or video. Let me know if you are interested in this option.

We would welcome the submission of a revised version within three months for further consideration. Please let us know if you require longer to complete the revision.

I look forward to receiving your revised manuscript.

Yours sincerely,

Zeljko Durdevic

We require:

- 1) A .docx formatted version of the manuscript text (including legends for main figures, EV figures and tables). Please make sure that the changes are highlighted to be clearly visible.
- 2) Individual production quality figure files as .eps, .tif, .jpg (one file per figure). For guidance, download the 'Figure Guide PDF': (<https://www.embopress.org/page/journal/17574684/authorguide#figureformat>).
- 3) A .docx formatted letter INCLUDING the reviewers' reports and your detailed point-by-point responses to their comments. As part of the EMBO Press transparent editorial process, the point-by-point response is part of the Review Process File (RPF), which will be published alongside your paper.
- 4) A complete author checklist, which you can download from our author guidelines (<https://www.embopress.org/page/journal/17574684/authorguide#submissionofrevisions>). Please insert information in the checklist that is also reflected in the manuscript. The completed author checklist will also be part of the RPF.
- 5) Please note that all corresponding authors are required to supply an ORCID ID for their name upon submission of a revised manuscript.
- 6) It is mandatory to include a 'Data Availability' section after the Materials and Methods. Before submitting your revision, primary datasets produced in this study need to be deposited in an appropriate public database, and the accession numbers and

database listed under 'Data Availability'. Please remember to provide a reviewer password if the datasets are not yet public (see <https://www.embopress.org/page/journal/17574684/authorguide#dataavailability>).

12) Author contributions: You will be asked to provide CRediT (Contributor Role Taxonomy) terms in the submission system. These replace a narrative author contribution section in the manuscript.

13) A Conflict of Interest statement should be provided in the main text.

14) Every published paper now includes a 'Synopsis' to further enhance discoverability. Synopses are displayed on the journal webpage and are freely accessible to all readers. They include a short stand first (maximum of 300 characters, including space) as well as 2-5 one-sentences bullet points that summarizes the paper. Please write the bullet points to summarize the key NEW findings. They should be designed to be complementary to the abstract - i.e. not repeat the same text. We encourage inclusion of key acronyms and quantitative information (maximum of 30 words / bullet point). Please use the passive voice. Please attach these in a separate file or send them by email, we will incorporate them accordingly.

15) Include a Reagents and Tools Table as part of the Methods section, which can be downloaded from our author guidelines (<https://www.embopress.org/page/journal/17574684/authorguide#structuredmethods>)

***** Reviewer's comments *****

Referee #1 (Remarks for Author):

The manuscript by Weidele et al. reports the development of micrometastasis-derived models for drug testing in high-risk, early-stage melanoma patients to improve treatment decisions and outcomes. The authors successfully generated patient-derived models from disseminated cancer cells (DCCs) obtained from the lymph nodes of melanoma patients, enabling functional drug testing prior to clinical relapse. Disseminated cancer cell density (DCCD) was identified as a significant predictor of successful model generation and patient outcome, with higher DCCD correlating with increased risk of death. The generated patient-derived xenograft (PDX) models retained histomorphological and some genomic characteristics similar to patient samples, demonstrating genomic stability over multiple passages. A drug screening of 315 anti-cancer drugs identified specific vulnerabilities in a melanoma-derived cell line, providing potential therapeutic options. DCC-derived models enabled personalised testing of responses to targeted treatments, identifying patients sensitive to therapies. Notably, the study demonstrated that patient-derived models could be generated in a timely manner, 82% of the cases prior to clinical progression, enabling personalised drug testing. Overall, the manuscript is well written and most experiments were carefully controlled. The conclusions are mostly supported by the data presented. However, there are several concerns that need to be addressed before the manuscript could be accepted for publication, as follows:

- To ensure that the models retained the genomics, genome-wide copy number alterations (CNA) were compared in this study. However, the comparison is qualitative and not quantitative, as shown in Fig. 4B,D and Suppl. Fig. 4. Unsupervised hierarchical clustering of all CNAs would be required to ensure that the samples corresponding to each patient are clustered together and not mixed with other patients.
- The comparison between samples/patients of other genomic parameters such as transcriptomics and whole-exome/genome mutations retained over multiple passages is missing in this study. If not performed, it should at least be described as a limitation in the discussion where only CNAs were considered.
- Although targeted panel sequencing identified the BRAF V600E mutation as oncogenic, it should be highlighted in the discussion that exome or whole genome sequencing comparing parental Mel-DCC-11 and BRAF-resistant Mel-DCC-11-R would provide a better picture.
- The samples of cell lines obtained both from PDXs (Approach I) and directly from patients (Approach II) should be analysed more carefully to understand the similarities and differences.
- The compound screening with 315 anticancer drugs was only performed on one CL model. Thus, this proof of concept is informative, but it should be emphasised in the text that while some mechanisms may be general to melanoma, certain drugs may be sensitive to patient Mel-53 but not others simply because of the inherent genetics/epigenetics of each sample.
- The authors should explain whether it would be possible to obtain enough cells to test 315 anti-cancer drugs in any of the successful models or discuss this limitation.
- Immune checkpoint inhibitors are the standard of care for adjuvant systemic therapy in Stage III and Stage IV (metastatic) melanoma, along with targeted therapies for BRAF V600E/K mutations. It appears that none of the 315 anticancer drugs are immune checkpoint inhibitors. This limitation should be discussed in the manuscript.
- In order to identify further associations, this study should include the immune checkpoint inhibitor or targeted therapy treatments received by the patients who developed metastatic melanoma.

Referee #2 (Remarks for Author):

In this interesting study, authors develop a collection of patient-derived xenograft and cell line models from disseminated melanoma cells in lymph nodes (LN). Authors characterise the models and link successful generation to disseminated cancer cell (DCC) quantity, LN origin (sentinel vs non-sentinel) and mortality risk. Most models were available before patient death or relapse, which enabled screening of anti-cancer drugs that could inform second-line treatment options. Authors propose that this protocol could select melanoma patients at high risk but with no overt metastatic disease, and who later could benefit from guided treatment decisions upon relapse.

Authors have carried out an impressive amount of work, using samples from >30 patients to generate DCC-derived PDXs from roughly half of the patient samples, and 13 PDX-derived cell lines. In other approach (II) authors established cell lines directly from dissociated cells or spheres. These models undoubtedly will be very useful for the melanoma research community.

My main concern is that despite generating PDX models, the drug screens are performed using cell lines, most likely for simplicity and lower costs, which is reasonable. However, did authors carry out drug screens on cell lines established directly in culture from the dissociated cells/spheres? Were results similar? If so, could PDX generation be skipped for some or most cases so screens can be performed faster?

Related to this, the origin of the cell line used in drug screens (Mel-DCC-11) is confusing, table 3 states for this cell line "origin PDX, LN-DCC" (See also related comment 4 below). Are there 2 Mel-DCC-11 lines, one established from PDX and one from LN-DCC? If so, which one was used for the drug screens?

Also, given the focus on Mel-DCC-11, which was derived from Patient 53, similar characterization of this patient and derived cell lines should be shown in Figure 2 -instead of in figure S3.

In the end, it is a bit unclear what the proposed pipeline is. Including a summary would help having the message come across clearer. Regarding the DCCD threshold of 100, Figure 5E convincingly shows reduced survival of patients with DCCD \geq 100 vs patients with no DCCD or <100, supporting author's statement "This observation supports the use of the DCCD {greater than or equal to} 100 threshold as a criterion for selecting patients for model generation at the time of primary treatment." However, there are a significant proportion of melanomas with DCCD \geq 100 that do not develop PDX (Figure 1B). So perhaps for deciding whether to follow the PDX route, other parameters should be included here, for example LN type (according to Figure 1E)?

Besides, there are a few issues that should be clarified.

1. How is the success in PDX generation measured? Is it just tumours that grew up to a certain volume? Is this the "maximal tumor burden" as stated in methods?

In any case, authors are encouraged to include representative tumour volume curves for select (or all) PDXs.

2. Disseminated cells in LNs are counted by labelling with gp-100 (HMB45). Then these LN-DCCs are expanded by xenotransplantation in mice (PDX) or through enrichment in melanospheres. In the PDX scenario, DCC are isolated from LNs using MCSP. Why was the melanoma marker changed here?

Including the markers used in the different isolations in Figure 1A would be useful.

3. Figure 2 and S3 show gp100 stainings on patient LN and PDXs, 1 picture per sample. Methods state that these stainings were evaluated by a pathologist. Some sort of quantification, qualitative or even better quantitative, should be provided.

4. Regarding cell line identification.

4a. Table 2. for cases in which there is CL from PDX and CL w/o PDX (CLs Mel-DCC-5a, 5b, 06, 07, 11), is the CL used later the one from PDX or the one not from PDX?

For example, for some in Table 3 (05, 06, 07, 10, 11), origin says PDX, LN-DCC.

4b. Figure S6 shows drug sensitivity for cell lines Mel-DCC-03, Mel-DCC-05 and DCC-10. Table 2 shows 3 different lines for DCC-5 and DCC-10 (a,b,c). Table 3 uses an "a" superscripted for both DCC-5 and 10 (comment below table says: "a" successful model establishment from different LNs of the same patient). So which cell lines were characterized in Table 3? And which cell lines were used in Figure S6?

5. This is the authors' choice and they may have their reasons for this, but the fact that the cell line does not have the same number as the patient from which it is established is confusing. For example, Mel-DCC-11 could have been Mel-DCC-53.

Referee #3 (Comments on Novelty/Model System for Author):

NSG mice models were used in this study which eliminates the impact of the immune system on any of these results and we know that the immune system definitely has an impact on drug sensitivity/ resistance

Referee #3 (Remarks for Author):

Why did the authors choose 120h for drug viability assay? What is the IC50 of the drugs used for the cell lines in the study?

In the table 3 with DCC cell lines, it would be helpful to note the IC50 of the sensitive cell line as a comparison in the same table

In this study, the authors are looking at acquired resistance- what about intrinsic resistance?

Studies like (PMID: 33082316/ 31506424) have attributed other RTKs like IRS-1 and MET as contributory factors to drug resistance. The authors should cite these articles and comment whether they have seen these come up in their study

“Micrometastasis-derived models enable drug testing for early-stage, high-risk melanoma patients” (EMM-2025-21888)**Point-by-point reply****Referee #1 (Remarks for Author):**

The manuscript by Weidele et al. reports the development of micrometastasis-derived models for drug testing in high-risk, early-stage melanoma patients to improve treatment decisions and outcomes. The authors successfully generated patient-derived models from disseminated cancer cells (DCCs) obtained from the lymph nodes of melanoma patients, enabling functional drug testing prior to clinical relapse. Disseminated cancer cell density (DCCD) was identified as a significant predictor of successful model generation and patient outcome, with higher DCCD correlating with increased risk of death. The generated patient-derived xenograft (PDX) models retained histomorphological and some genomic characteristics similar to patient samples, demonstrating genomic stability over multiple passages. A drug screening of 315 anti-cancer drugs identified specific vulnerabilities in a melanoma-derived cell line, providing potential therapeutic options. DCC-derived models enabled personalised testing of responses to targeted treatments, identifying patients sensitive to therapies. Notably, the study demonstrated that patient-derived models could be generated in a timely manner, 82% of the cases prior to clinical progression, enabling personalised drug testing. Overall, the manuscript is well written and most experiments were carefully controlled. The conclusions are mostly supported by the data presented. However, there are several concerns that need to be addressed before the manuscript could be accepted for publication, as follows:

- To ensure that the models retained the genomics, genome-wide copy number alterations (CNA) were compared in this study. However, the comparison is qualitative and not quantitative, as shown in Fig. 4B,D and Suppl. Fig. 4. Unsupervised hierarchical clustering of all CNAs would be required to ensure that the samples corresponding to each patient are clustered together and not mixed with other patients.

Reply: As suggested by the reviewer, we analyzed, whenever possible, patient samples, *ex vivo* cultured material and the generated models by fingerprint analysis (STR) and copy number alterations (CNAs). We now included unsupervised clustering of the STR data in Figure EV1A and the unsupervised hierarchical clustering of the CNA data (Figure EV3A) and added appropriate description as requested. STR and CNA clustering confirm that the samples correspond to the appropriate patients and that the alterations acquired during natural cancer evolution were not overridden by *in vitro* selection.

“STR profile of PDX samples and CLs were compared to the profile of the corresponding DCC. All analysed patient samples matched each other”; page 19

“In all tested cases, CNA profiles of matching samples were highly congruent also being confirmed by hierarchical clustering”; page 8

- The comparison between samples/patients of other genomic parameters such as transcriptomics and whole-exome/genome mutations retained over multiple passages is

missing in this study. If not performed, it should at least be described as a limitation in the discussion where only CNAs were considered.

Reply: Indeed, in this study we focused on analyzing copy number alterations (CNAs) among our samples and passages (as shown in Figure EV3) without including other genomic parameters such as transcriptomics and whole exome/genome mutations. We consider a comparison of transcriptomes without clear question as difficult to interpret, since cellular plasticity and response to culture conditions will impact and alter transcriptomes. WES or WGS data were not addressed in this study as costs and effort go beyond the scope of this paper. Therefore, as suggested, we added to the discussion: "To better understand the DCC genotype and phenotype of Mel-DCC CLs at different passages and various culture conditions, in-depth analyses such as whole exome/genome sequencing and transcriptome analyses will be required.", page 14

- Although targeted panel sequencing identified the BRAF V600E mutation as oncogenic, it should be highlighted in the discussion that exome or whole genome sequencing comparing parental Mel-DCC-11 and BRAF-resistant Mel-DCC-11-R would provide a better picture.

Reply: As suggested, we included in the discussion that the implementation of whole exome or genome sequencing approaches could offer deeper insights: "Additionally, the application of whole exome or genome sequencing approaches will provide a more comprehensive understanding of how intrinsic or acquired drug resistance to targeted therapies is caused and can be effectively addressed."; page 14

- The samples of cell lines obtained both from PDXs (Approach I) and directly from patients (Approach II) should be analyzed more carefully to understand the similarities and differences.

Reply: To gain a first glimpse into similarities or differences, we analyzed the melanoma (-associated) cell surface marker expression profile by flow cytometry of the CLs, which could be generated via expansion approach I and II (derived from patient Mel-38, -40, -45, -51 and -53). We found a comparable expression of MCSP, CD63, CD146, CD166 and CD271 in cell lines generated via approach I vs. II.

We furthermore investigated the genomic plasticity of CLs generated using both approaches by examining individual isolated cells using low-pass whole-genome sequencing. We compared single cells from CLs derived from patient Mel-53 and generated both via expansion approach I ('PDX', n = 13 single cells) and approach II ('direct', n = 13 single cells). When applying hierarchical clustering, we did not observe any separation between the cells of the CLs generated by approach I or II. This indicates genomic heterogeneity, which is comparably present in both approaches, but also shows a high similarity between the two approaches. Based on this data no selection of specific genomic clones could be observed based on the expansion approaches. However, deeper analysis such as WXS might discover more differences.

Both analyses are now included into the updated manuscript in Appendix Figure S3 and added to the results part "Cell lines generated in parallel by approaches I and II displayed similar key features, including melanoma surface-marker expression and CNA profiles"; page 8

- The compound screening with 315 anticancer drugs was only performed on one CL model. Thus, this proof of concept is informative, but it should be emphasised in the text that while some mechanisms may be general to melanoma, certain drugs may be sensitive to patient Mel-53 but not others simply because of the inherent genetics/epigenetics of each sample.

Reply: We fully agree. The generation of patient-derived cell lines, such as Mel-DCC-11, is intended to enable personalized drug testing. As drug responses can be influenced by the unique genetics and epigenetics of each tumor—as well as the individual therapeutic history of the patient—our approach aims to capture this complexity and variability. Since the current screen serves as a proof of concept, we have clarified in the manuscript that certain drug sensitivities observed may be specific to Mel-53 and not broadly generalizable to all melanoma cases. “The observed drug sensitivities are likely to be patient specific and not broadly generalizable to all melanoma cases, given the unique genetic and epigenetic landscapes of each tumor and the patient’s individual treatment history”; page 14

- The authors should explain whether it would be possible to obtain enough cells to test 315 anti-cancer drugs in any of the successful models or discuss this limitation.

Reply: The generated cell lines can be propagated and passaged like commercial cell lines. Consequently, it is possible to obtain enough cells to test 315 anti-cancer drugs in any of the successful models. As exemplified for Mel-DCC-11, the CL models offer the potential for in-depth analyses, including large scale drug screening approaches for drug discovery or additional molecular analyses. We addressed this now in the discussion: "...they offer versatile tools for in-depth analyses, such as large-scale drug screenings, enabling the exploration of resistance mechanisms to guide the development of more effective therapeutic strategies"; page 14

- Immune checkpoint inhibitors are the standard of care for adjuvant systemic therapy in Stage III and Stage IV (metastatic) melanoma, along with targeted therapies for BRAF V600E/K mutations. It appears that none of the 315 anticancer drugs are immune checkpoint inhibitors. This limitation should be discussed in the manuscript.

Reply: We agree. The omission of immune checkpoint inhibitors (ICIs) in the original drug screen was due to the fact that ICIs efficacy could not be evaluated because of the absence of immune cells in our drug screening assay. However, to address this limitation and to demonstrate the feasibility of ICI testing using our CL models, **we conducted additional proof-of-concept coculture experiments including treatment of ICIs. This has now been added as a new Figure 5.**

- In order to identify further associations, this study should include the immune checkpoint inhibitor or targeted therapy treatments received by the patients who developed metastatic melanoma.

Reply: Our initial manuscript mainly focused on targeted therapies. As the reviewer points out immune checkpoint inhibitors have become the most important component in the treatment of melanoma and should therefore be included in *in vitro* testing. We needed to modify the assay and have **now integrated a coculture of Mel-DCC CLs and LN cells in**

Figure 5, which now allows us to study the interactions between the two cell types as well as effects of ICI treatment. Thus, the revised manuscript now demonstrates in a proof-of-concept that also immune therapies can be addressed in a timely manner.

Referee #2 (Remarks for Author):

In this interesting study, authors develop a collection of patient-derived xenograft and cell line models from disseminated melanoma cells in lymph nodes (LN). Authors characterise the models and link successful generation to disseminated cancer cell (DCC) quantity, LN origin (sentinel vs non-sentinel) and mortality risk. Most models were available before patient death or relapse, which enabled screening of anti-cancer drugs that could inform second-line treatment options. Authors propose that this protocol could select melanoma patients at high risk but with no overt metastatic disease, and who later could benefit from guided treatment decisions upon relapse.

Authors have carried out an impressive amount of work, using samples from >30 patients to generate DCC-derived PDXs from roughly half of the patient samples, and 13 PDX-derived cell lines. In other approach (II) authors established cell lines directly from dissociated cells or spheres. These models undoubtedly will be very useful for the melanoma research community.

My main concern is that despite generating PDX models, the drug screens are performed using cell lines, most likely for simplicity and lower costs, which is reasonable. However, did authors carry out drug screens on cell lines established directly in culture from the dissociated cells/spheres? Were results similar? If so, could PDX generation be skipped for some or most cases so screens can be performed faster?

Reply: *In vivo* drug screening was not the focus of this study due to the high administrative effort and costs associated with animal testing as well as the low throughput which goes along with longer periods for the *in vivo* drug screening process itself.

Regarding the comparability of CLs generated *in vivo* (approach I) vs. direct *in vitro* expansion (approach II), we have now added additional data. First, we phenotypically compared basic melanoma-associated surface marker expression (MCSP, CD63, CD145, CD166 and CD271) which showed a very comparable expression between CLs from both approaches (see new Appendix Figure S3A). Second, we performed single cell CNV sequencing of both CLs derived from patient Mel-53 (approach I, n = 13 single cells vs. approach II, n = 13 single cells). When applying hierarchical clustering, we did not observe any separation between the cells of the CLs generated by approach I or II. This indicates that based on the phenotypic and CNV characterization similar clones prevail *in vitro* and *in vivo* (Appendix Figure S3B).

In addition, we have used CLs generated by direct expansion (approach II) for screening the targeted therapies (Figure 4 A, B). We apologize that in our original manuscript, the origin of the CLs used for drug screening was not well described, and we have therefore adjusted this accordingly in Table 3 (see also next comment).

The perspective of even saving further time through the direct generation of cell lines without prior *in vivo* expansion, particularly in high-risk patients, has now been presented in more detail in the discussion; page 13.

Related to this, the origin of the cell line used in drug screens (Mel-DCC-11) is confusing, table 3 states for this cell line "origin PDX, LN-DCC" (See also related comment 4 below). Are there 2 Mel-DCC-11 lines, one established from PDX and one from LN-DCC? If so, which one was used for the drug screens?

Reply: We thank the reviewer for pointing out that this was not well described in our original manuscript. As outlined in the updated Table 2, we were able to successfully generate parallel cell lines from five patients via expansion approach I and II (from patient Mel-38, -40, -45, -51 and -53 (the source of Mel-DCC-11)). However, for a more detailed characterization, including drug testing, we concentrated on one model per patient. Since models that could be expanded directly (approach II, without xenograft) were slightly faster available in most of the cases, we focused on those models. These are now listed (including the clearly defined origin) in the updated Table 3.

Also, given the focus on Mel-DCC-11, which was derived from Patient 53, similar characterization of this patient and derived cell lines should be shown in Figure 2 -instead of in Figure S3.

Reply: As suggested, we included the data for patient Mel-53 and the derived models in Figure 2A.

In the end, it is a bit unclear what the proposed pipeline is. Including a summary would help having the message come across clearer. Regarding the DCCD threshold of 100, Figure 5E convincingly shows reduced survival of patients with $DCCD \geq 100$ vs patients with no DCCD or < 100 , supporting author's statement "This observation supports the use of the DCCD {greater than or equal to} 100 threshold as a criterion for selecting patients for model generation at the time of primary treatment." However, there are a significant proportion of melanomas with $DCCD \geq 100$ that do not develop PDX (Figure 1B). So perhaps for deciding whether to follow the PDX route, other parameters should be included here, for example LN type (according to Figure 1E)?

Reply: To address the valid point made by the reviewer, we added the following explanation to the discussion: "When establishing cell lines from patient-derived samples, use of NSLNs resulted in a higher success rate. This may be rooted in the fact that involvement of NSLN reflects very high DCCD values. However, the observation that melanoma cells from NSLNs displayed higher numbers of CNAs and that LN type was an independent factor in multivariable analysis may indicate further biological differences between SLN and NSLN-derived melanoma cells relevant for CL generation. Since CLs could be established directly from patient samples without prior xenotransplantation in five cases (one SLN, four NSLNs, all with $DCCD > 50,000$), direct CL generation can be recommended for high-risk patients with very high DCCD counts. However, if the goal is to maximize the chance of expanding DCCs even from patients with low DCCD, using a PDX will be useful despite substantial resources in terms of cost, logistics, personnel, and regulatory effort.", page 13

Besides, there are a few issues that should be clarified.

1. How is the success in PDX generation measured? Is it just tumours that grew up to a certain volume? Is this the "maximal tumor burden" as stated in methods? In any case, authors are encouraged to include representative tumour volume curves for select (or all) PDXs.

Reply: The aim of our study was to test whether DCCs grow *in vivo* at all and to propagate DCCs for *in vitro* cell line generation and drug screening. To harvest enough cells, the tumors were allowed to grow to the maximum tumor size (10 mm diameter), which was evaluated as 'success in PDX generation'. At 10 mm mice must be euthanized according to animal experimentation regulations. However, in case tumors showed ulceration or immunodeficient mice signs of sickness, mice were euthanized and tumors harvested before reaching the maximum size. As it was not our aim to compare growth curves, mice were only palpated weekly without documenting the tumor size. We now present this more clearly in the methods section. page 17

2. Disseminated cells in LNs are counted by labelling with gp-100 (HMB45). Then these LN-DCCs are expanded by xenotransplantation in mice (PDX) or through enrichment in melanospheres. In the PDX scenario, DCC are isolated from LNs using MCSP. Why was the melanoma marker changed here?

Including the markers used in the different isolations in Figure 1A would be useful.

Reply: We thank the reviewer for this comment. We used the well-established gp100 assay which allows the quantitative evaluation of the samples. However, gp100 as an intracellular marker precludes viable cell isolation for *in vivo/in vitro* expansion. Thus, for model development we identified potential LN-DCCs by the cell surface marker MCSP and transplanted these into mice.

As suggested, we have modified the Figure 1A and the corresponding figure legend, now providing additional information.

3. Figure 2 and S3 show gp100 stainings on patient LN and PDXs, 1 picture per sample. Methods state that these stainings were evaluated by a pathologist. Some sort of quantification, qualitative or even better quantitative, should be provided.

Reply: We updated the Methods section accordingly and added the H-scores to the respective IHC figures.

4. Regarding cell line identification.

4a. Table 2. for cases in which there is CL from PDX and CL w/o PDX (CLs Mel-DCC-5a, 5b, 06, 07, 11), is the CL used later the one from PDX or the one not from PDX?

For example, for some in Table 3 (05, 06, 07, 10, 11), origin says PDX, LN-DCC.

4b. Figure S6 shows drug sensitivity for cell lines Mel-DCC-03, Mel-DCC-05 and DCC-10. Table 2 shows 3 different lines for DCC-5 and DCC-10 (a,b,c). Table 3 uses an "a" superscripted for both DCC-5 and 10 (comment below table says: "a" successful model establishment from different LNs of the same patient). So which cell lines were characterized in Table 3? And which cell lines were used in Figure S6?

Reply: For clarity, we updated Table 2 and 3.

5. This is the authors' choice and they may have their reasons for this, but the fact that the cell line does not have the same number as the patient from which it is established is confusing. For example, Mel-DCC-11 could have been Mel-DCC-53.

Reply: We chose an CL ID independent of the patient number, based on the order of cell line generation/outgrowth. The Mel-DCC CLs here described are part of a larger collection of patient-derived CL models. Maintaining this unique identifier independent of the patient number will support easier cross-referencing.

Referee #3 (Comments on Novelty/Model System for Author):

NSG mice models were used in this study which eliminates the impact of the immune system on any of these results and we know that the immune system definitely has an impact on drug sensitivity/resistance

Reply: Thank you for the valuable comment. To overcome the limitation spotted by the reviewer, we have now established a new coculture model of lymph node cells and Mel-DCC CLs in Figure 5, which allows us to study the interactions between the two cell types as well as effects of ICI treatment. Unfortunately, due to the time limitation during the revision, no larger drug screens could be analyzed within this coculture model so far. However, these first results provide an important outlook for our future studies.

Referee #3 (Remarks for Author):

Why did the authors choose 120h for drug viability assay? What is the IC50 of the drugs used for the cell lines in the study?

Reply: Incubation times of 72 to 120 hours are commonly used to assess the in vitro drug efficacy of BRAF/MEK inhibitors in melanoma cell lines. Since the doubling times of our established cell line models ranging from ~42 h up to 119 h, we chose the incubation time of 120 h to provide an accurate assessment of drug efficacy by allowing sufficient time for the drug to exert its effects on the less frequently cycling cells. Please see Table 3 for IC50 values for Vemurafenib and Binimetinib.

In the table 3 with DCC cell lines, it would be helpful to note the IC50 of the sensitive cell line as a comparison in the same table

Reply: Since the IC50 values for Vemurafenib and Binimetinib for the patient-derived cell lines are listed in Table 3, we assume the reviewer is asking about the resistant models. In these models, the IC50 was not reached (drug concentration did not kill half the cells) in the range of the tested dose-response.

In this study, the authors are looking at acquired resistance- what about intrinsic resistance?

Reply: We also identified two cases of potential intrinsic resistance. CL models of these patients harbor mutations expecting potential responses to NRAS inhibition (Mel-DCC-04: NRAS Q61R and Mel-DCC-07: NRAS T58I). However, they responded only weakly to Binimetinib (Figure 4A), indicating a pre-existing drug resistance. That mechanism could be addressed by further investigations, however dissecting the underlying mechanism will require additional experiments beyond the scope of this paper.

In general, the whole workflow for personalized drug screening proposed in our manuscript intends to functionally identify drugs with high sensitivity but also drugs that do not work due to intrinsic resistance. The latter therefore would be extremely important information for individual patients to avoid treatments that obviously will not work.

Studies like (PMID: 33082316 / 31506424) have attributed other RTKs like IRS-1 and MET as contributory factors to drug resistance. The authors should cite these articles and comment whether they have seen these come up in their study

Reply: We appreciate the reviewer's insightful comment. To answer this question, we took a closer look at the drug screening data performed on the parental Vemurafenib sensitive and resistant CL model Mel-DCC-11. The tested library contained 3 out of 315 drugs which are assigned to target c-Met (or MET), namely Crizotinib, Cabozantinib and the s-malate salt form of Cabozantinib. As depicted in Figure 4D, Cabozantinib and the s-malate salt form of Cabozantinib, both strongly affected the viability of resistant Mel-DCC-11 cells in presence of Vemurafenib. We now have emphasized c-Met in the text passage in the results section accordingly.

To our knowledge, there are up to date no FDA-approved anti-cancer drugs specifically and solely targeting IRS-1 that were consequently not present in the library we used in our study. However, some drugs targeting related pathways like PI3K and mTOR, which are part of the IRS-1 signaling network, were identified to effectively kill the Mel-DCC11 R+V cells (Figure 4D: Copanlisib, Temsirolimus, Rapamycin).

In this context, we cited publications from Dratkiewicz et al. (Characterization of Melanoma Cell Lines Resistant to Vemurafenib and Evaluation of Their Responsiveness to EGFR- and MET-Inhibitor Treatment) and Wang et al (Targeting mTOR signaling overcomes acquired resistance to combined BRAF and MEK inhibition in BRAF-mutant melanoma).

21st Oct 2025

Dear Dr. Klein,

Thank you for the submission of your revised manuscript to EMBO Molecular Medicine. I am pleased to inform you that we will be able to accept your manuscript pending the following final amendments:

- 1) Please address referee #2 minor point.
- 2) Figures:
 - Please reduce Figure 2 and Figure EV3 file size.
 - We note that in Appendix Figure S1 images Mel-09 Patient and PDX appear to be identical. Please clarify and correct.
- 3) Authors: E-mail correspondence to Melanie Werner-Klein and Marie Katzer could not be delivered. Please update their e-mail addresses and make sure to enter correct e-mail addresses for all authors in our submission system.
- 4) In the main manuscript file, please do the following:
 - Please address all comments suggested by our data editors listed below:
 - o Data availability statement:
 1. Please note that the specific URL for EGAS50000001225 dataset is not provided in the data availability statement.
 - o Figure legends:
 1. Please note that the exact p values are not provided in the legends of figures 1B, 6E.
 - In Methods, provide the statement that informed consent was obtained from all human subjects and that the experiments conformed to the principles set out in the WMA Declaration of Helsinki and the Department of Health and Human Services Belmont Report.
 - Rename "Competing interests statement" to "Disclosure and competing interests statement". We updated our journal's competing interests policy in January 2022 and request authors to consider both actual and perceived competing interests. Please review the policy <https://www.embopress.org/competing-interests> and update your competing interests if necessary.
 - Indicate in legends exact n and exact p values, not a range, along with the statistical test used. To keep the figures "clear" some authors found providing an Appendix table Sx with all exact p-values preferable. You are welcome to do this if you want to.
 - In data availability statement please remove the sentence "All other data that support the findings of this study are available within the article, its Extended Data, and Appendix, or from the corresponding author upon reasonable request." Please use the following format to report the accession number of your data:

[data type]: [full name of the resource] [accession number/identifier] ([doi or URL or identifiers.org/DATABASE:ACCESSION])

Please check "Author Guidelines" for more information.

<https://www.embopress.org/page/journal/17574684/authorguide#availabilityofpublishedmaterial>

- Correct the reference citation in the reference list. Where there are more than 10 authors on a paper, 10 will be listed, followed by "et al.". Also, please remove DOIs. Please check "Author Guidelines" for more information.

<https://www.embopress.org/page/journal/17574684/authorguide#referencesformat>

5) Tables: Please add a legend to Table EV1.

6) Appendix: Please add page numbers to the table of contents and reduce the size of the file.

7) Synopsis:

8) As part of the EMBO Publications transparent editorial process initiative (see our Editorial at <http://embomolmed.embopress.org/content/2/9/329>), EMBO Molecular Medicine will publish online a Review Process File (RPF) to accompany accepted manuscripts. This file will be published in conjunction with your paper and will include the anonymous referee reports, your point-by-point response and all pertinent correspondence relating to the manuscript. Let us know whether you agree with the publication of the RPF and as here, if you want to remove or not any figures from it prior to publication. Please note that the Authors checklist will be published at the end of the RPF.

9) Please provide a point-by-point letter INCLUDING my comments as well as the reviewer's reports and your detailed responses (as Word file).

I look forward to reading a new revised version of your manuscript as soon as possible.

Yours sincerely,

Zeljko Durdevic

Zeljko Durdevic

*** Instructions to submit your revised manuscript ***

- 1) a .docx formatted version of the manuscript text (including Figure legends and tables)
 - 2) Separate figure files*
 - 3) supplemental information as Expanded View and/or Appendix. Please carefully check the authors guidelines for formatting Expanded view and Appendix figures and tables at <https://www.embopress.org/page/journal/17574684/authorguide#expandedview>
 - 4) a letter INCLUDING the reviewer's reports and your detailed responses to their comments (as Word file).
 - 5) The paper explained: EMBO Molecular Medicine articles are accompanied by a summary of the articles to emphasize the major findings in the paper and their medical implications for the non-specialist reader. Please provide a draft summary of your article highlighting
 - the medical issue you are addressing,
 - the results obtained and
 - their clinical impact.This may be edited to ensure that readers understand the significance and context of the research. Please refer to any of our published articles for an example.
 - 6) Author contributions: the contribution of every author must be detailed in a separate section.
 - 7) EMBO Molecular Medicine now requires a complete author checklist (<https://www.embopress.org/page/journal/17574684/authorguide>) to be submitted with all revised manuscripts. Please use the checklist as guideline for the sort of information we need WITHIN the manuscript. The checklist should only be filled with page numbers where the information can be found. This is particularly important for animal reporting, antibody dilutions (missing) and exact values and n that should be indicated instead of a range.
 - 8) Every published paper now includes a 'Synopsis' to further enhance discoverability. Synopses are displayed on the journal webpage and are freely accessible to all readers. They include a short stand first (maximum of 300 characters, including space) as well as 2-5 one sentence bullet points that summarise the paper. Please write the bullet points to summarise the key NEW findings. They should be designed to be complementary to the abstract - i.e. not repeat the same text. We encourage inclusion of key acronyms and quantitative information (maximum of 30 words / bullet point). Please use the passive voice. Please attach these in a separate file or send them by email, we will incorporate them accordingly.
- You are also welcome to suggest a striking image or visual abstract to illustrate your article. If you do please provide a jpeg file 550 px-wide x 300-600px high.
- 9) A Conflict of Interest statement should be provided in the main text
 - 10) Please note that we now mandate that all corresponding authors list an ORCID digital identifier. This takes <90 seconds to

complete. We encourage all authors to supply an ORCID identifier, which will be linked to their name for unambiguous name identification.

Currently, our records indicate that there is no ORCID associated with your account.

Please click the link below to provide an ORCID:

Link Not Available

11) Include a Reagents and Tools Table as part of the Methods section, which can be downloaded from our author guidelines (<https://www.embopress.org/page/journal/17574684/authorguide#structuredmethods>)

Photos 400-800 DPI

*Additional important information regarding figures and illustrations can be found at

<https://bit.ly/EMBOPressFigurePreparationGuideline>. See also figure legend preparation guidelines:

<https://www.embopress.org/page/journal/17574684/authorguide#figureformat>

***** Reviewer's comments *****

Referee #1 (Remarks for Author):

The authors successfully addressed the most critical concerns raised by this reviewer. The proof-of-concept co-culture experiments for testing immune checkpoint inhibitors could greatly enhance the usefulness of the patient-derived melanoma models described in this study. The manuscript is therefore now suitable for publication.

Referee #2 (Remarks for Author):

I thank the authors for addressing my comments and their clarifications. This study will be a valuable resource for the scientific community. I support publication.

One minor comment: for Appendix Figure S3A, the color scale does not have a color gradient, it is just one shade of blue, while the date on the left does show different shades of blue and whites (for very low/close to zero values).

The authors addressed the remaining formatting issues.

30th Oct 2025

Dear Dr. Klein,

We are pleased to inform you that your manuscript is accepted for publication and is now being sent to our publisher to be included in the next available issue of EMBO Molecular Medicine.

Zeljko Durdevic
Senior Editor
EMBO Molecular Medicine
